# Developing a pathway-independent and full-autonomous global resource allocation strategy to dynamically switching phenotypic states

Junjun Wu [1✉], Meijiao Bao[1], Xuguo Duan [2], Peng Zhou[1], Caiwen Chen[1], Jiahua Gao[1], Shiyao Cheng[1], Qianqian Zhuang[3] & Zhijun Zhao[4]

A grand challenge of biological chemical production is the competition between synthetic circuits and host genes for limited cellular resources. Quorum sensing (QS)-based dynamic pathway regulations provide a pathway-independent way to rebalance metabolic flux over the course of the fermentation. Most cases, however, these pathway-independent strategies only have capacity for a single QS circuit functional in one cell. Furthermore, current dynamic regulations mainly provide localized control of metabolic flux. Here, with the aid of engineering synthetic orthogonal quorum-related circuits and global mRNA decay, we report a pathway-independent dynamic resource allocation strategy, which allows us to independently controlling two different phenotypic states to globally redistribute cellular resources toward synthetic circuits. The strategy which could pathway-independently and globally self-regulate two desired cell phenotypes including growth and production phenotypes could totally eliminate the need for human supervision of the entire fermentation.

[1] College of Food Science and Technology, Nanjing Agricultural University, Nanjing, Jiangsu 210095, China. [2] Department of Food Science and Technology, College of Light Industry and Food Engineering, Nanjing Forestry University, Nanjing 210037, China. [3] State Key Laboratory of Biobased Material and Green Papermaking, School of Bioengineering, Qilu University of Technology, Jinan 250353, China. [4] Biorefinery Laboratory, Shanghai Advanced Research Institute, Chinese Academy of Sciences, 99 Haike Road, Shanghai 201210, China. ✉email: wujunjun@njau.edu.cn

A fundamental goal of engineering microbes for renewable synthesis of value-added chemicals from low-cost substrates is to maximizing their productivities, titers, and yields to realize economic feasibility[1]. However, the unforeseen competition between cellular metabolism and target synthetic circuit over shared cellular resources included metabolites, amino acids, nucleotides, ribosomes, and transfer RNAs often arises during bioprocesses, thus limiting final product yields and titers[2]. Traditional genetic modifications (i.e., gene overexpression and knockout) to alleviate this competition are typically static, and cannot address such challenges, especially for complex pathways and products which need to monitor environmental conditions and respond to adjust metabolic behavior during the production course[3]. For instances, knocking out a gene often divert cellular resources too heavily away from natural pathways due to its permanent and continuous impact especially for cell essential genes, resulting in cell death especially in cases of essential genes[4].

Compared to static regulation, dynamic control is prevalent in organisms by nature and helps cells to adapt organismal metabolic states to environmental changes in real time, which can self-regulate and redistribute cellular pathway fluxes to maximize product yields and minimize human supervision of fermentation process control[5]. Less external intervention is absolutely necessary if cells are re-programmed to undergo multiple steps during bioprocesses. Several exciting researches have already demonstrated the successful implementation of dynamic pathway regulation in academic metabolic engineering projects. For example, promoter-regulator systems detecting acetyl phosphate[6], FadR-based sensors[7], stress-responsive promoters sensing farnesyl pyrophosphate[8], malonyl-CoA sensor-regulators, and bi-functional dynamic control regulating phosphoenolpyruvate metabolic nodes[2] have been designed to dynamically control the biosynthesis of lycopene, biodiesel, amorphadiene, fatty acids and muconic acid. While these exciting achievements have been proven effective, such strategies require pathway- or metabolite-specific sensors, which are unknown for many metabolites or pathways that one might want to monitor, thus limiting their widespread use in other metabolic pathways or hosts[9].

As such, developing broadly applicable genetic circuits that could be portable across contexts is highly desirable to extend dynamic regulation to diverse synthetic pathways. Additionally, inducible promoters, although valuable for driving gene expression toward target synthetics circuits from research purpose standpoints, are not economically feasible for an industrial setting due to their high expenses and additional costs in downstream processing[3,9]. Previous pioneering work showed that quorum-sensing (QS) system, which functions to control cell population-dependent behaviors in bacteria, could be applied to create a pathway- or metabolite-independent genetic device for dynamical gene expression control[3,10]. Prather and coworkers elegantly rewired Esa QS system from *Pantoea stewartia* to redirect glycolytic flux into target synthetic circuits, significantly advancing our understanding of implementing QS-based dynamic regulation system in cellular setting[3].

However, in most cases, these pathway-independent strategies for dynamically regulating pathway fluxes are often utilizing a single QS circuit for inhibiting competing pathways and need another inducible promoter for delaying target pathway expression. In these cases, one cell only allow for a single QS circuit fully functional for up-regulation. These semi-autonomous modes still rely on inducer supplementation and cannot eliminate human supervision during the course of fermentation[3]. Furthermore, current dynamic regulation mainly provides localized control of metabolic flux, and it does not redistribute the global allocation of cellular subsystems and states[3]. Hence, technologies should be developed to full-autonomously, genome-widely regulate gene expression patterns to minimize resource expenditure.

Here, a dynamic, pathway-independent and full-autonomous global resource allocation strategy was developed to solve these challenges (Supplementary Fig. 1). After evaluating numerous quorum sensing (QS) systems from both Gram-negative and -positive bacterial species, the peptide pheromone responsive QS system from *Enterococcus faecalis* and acyl-homoserine lactone responsive QS system from *Vibrio fischeri* were identified to exhibit high dynamic range and completely no crosstalk for each other when co-expressed in one cell. Moreover, we found that both QS systems remained a residual activity in the absence of signaling molecules, which would hinder the ability to maintain a gene in the off state, especially for toxic proteins, and engineering promoter sequence and protein regulator would dramatically decrease their leaky expression. These two orthogonal, autonomous and tunable QS-based circuits were then layered to dynamically modulate expression of two different sets of enzymes in the same host. Furthermore, the sequence-dependent endoribonuclease MazF was combinatorially used with QS-based circuits to build a global resource allocation device. As a demonstration, this dynamic resource allocation controller was implemented to control the production of medium chain fatty acids (MCFAs) in both shake flask cultures and 5-L bioreactors, exhibiting the potential of large-scale application.

## Results

### Characterization of QS systems from both Gram-negative and -positive bacterial species.
To establish a full autonomous, pathway-independent genetic device dynamically controlling endogenous gene expression and synthetic circuits simultaneously, two QS systems completely orthogonal in operation should be identified. Although there are numerous pioneering studies characterizing different QS systems to investigate orthogonal pairs[11–18], the available QS systems that could be functional in a single cell without cross talk were still limited. Previous studies mainly focused on characterizing Lux-like QS systems from Gram-negative bacterium[11,12], each with a unique LuxR-like receptor and homoserine lactone (HSL) homologue, and found that hybrid tra and rpa QS systems, which were created by replacing lux-box-like sequences from Lux QS systems with tra-box and rpa-box, respectively, exhibited both signal and promoter orthogonality[12]. However, this hybrid characteristic often led to off-target binding of regulator proteins toward motifs of lux box promoter when co-expressed in a single cell[11].

Here, in order to identify orthogonal QS systems exhibiting low cross talk and high dynamic range, which are two important parameters of sensors, we cloned several QS systems from both Gram-negative and -positive species based on ligand uniqueness, including *Vibrio fischeri*, *Pseudomonas aeruginosa*, *Staphylococcus aureus*, and *Enterococcus faecalis*, to expand the screen range. In most cases, Gram-negative bacteria utilize homoserine lactones as signaling ligands, whereas Gram-positive bacteria employ small peptides[19]. The major differences in signaling ligands indicate the potential of identifying orthogonal QS systems functional in a single cell.

In Gram-negative bacteria, membrane-permeable compound HSL-based QS systems had become most popular tools for constructing gene regulators. We chose Lux system from *V. fischeri* and Las system from *P. aeruginosa*, as these two HSL-based systems had been extensively characterized, and many other HSL-based systems did not exhibit response function in the presence of cognate receptor and ligand[12]. The *V. fischeri* QS system includes LuxI (ligand synthase) and LuxR (transcription activator), and these two regulatory genes would bind to each other to activate the Lux operon promoter (PluxI)[20]. The *P. aeruginosa* QS system consists of auto-inducer synthase LasI and transcriptional activator LasR, both of which are required for the expression of virulence

factors, such as elastase LasB, protease LasA[21], and hydrogen cyanide synthase hcnABC[22].

Gram-positive bacteria have two types of QS system including one-component (using cytoplasmic peptide receptors) and two-component (using membrane-bound receptors) QS systems[19]. However, to the best of our knowledge, few studies have characterized the full response function of Gram-positive QS systems in E. coli, and most of them need additional supplementation of auto-inducers to activate gene circuits fully[17]. Here, the agr two-component QS system of S. aureus and prgX one-component QS system of E. faecalis were chosen for charactering in E. coli. The agr system consists of P2 and P3 promoters upstream of RNAII and RNAIII transcripts, respectively. The RNAII transcript includes propeptide synthase AgrD, transmembrane endopeptidase AgrB, transmembrane histidine kinase receptor AgrC, and transcriptional activator AgrA. AgrD and AgrB are responsible for synthesizing mature auto-inducing signal peptide, and AgrA and AgrC constitute signal transduction system to activate P3 promoter[23]. In PrgX QS system, the master protein regulator PrgX binds the operator sequence of the response promoter from prgQ operon and represses the expression of this operon. Once bound to heptapeptide cCF10 produced by heptapeptide synthase CcfA, a conformational change would occur to the protein regulator PrgX and could no longer interact with the operator sequence, thus led to the activation of prgQ operon[24,25].

To facilitate examining the interactions of different QS components, a modular three-plasmid system constructed in our previous studies, which could be stably maintaining together in each cell due to their different origins and resistance marker[26], is used. Such that a signal plasmid (Ps) and a regulator plasmid (Pr) produced auto-inducers and transcription regulators under their native promoters, respectively, while a reportorial plasmid expressed GFP under a single QS-promoter (Pp) (Fig. 1a). All fluorescence measurements are normalizing by dividing measured fluorescence values by the $OD_{600}$ of that well to conduct a per-cell measurement. The function of each QS system is validated by comparing normalized GFP fluorescence from strains harboring plasmids of Ps, Pa, and Pp to strains only harboring Pp and corresponding empty plasmids.

The function of each QS system was validating by dynamic range defined as the ratio of comparing GFP fluorescence from strains harboring plasmids of Ps, Pr, and Pp to strains only harboring Pp. As seen from Fig. 1b, it was found that different QS systems exhibited significant varying response function. For three QS systems from P. aeruginosa, QS systems harboring LasB response promoter (QPB) and LasA response promoter (QPA) exhibited significant higher GFP induction expression than harboring HcnABC promoter (QPH). For QS systems from Gram-negative bacteria, Lux system from V. fischeri (QVX) exhibited higher GFP induction expression than all the Las systems from P. aeruginosa. However, none of the QS systems from Gram-positive bacteria showed GFP induction. In order to figure out the possible points of failure, the QS components, such as signal and activator synthases were GFP tagged to measure translational output directly (Fig. 1c). We found that QS component-fusions from Gram-negative bacteria were shown to express well, while fusions from Gram-positive bacteria exhibited non-expression. It was speculated that native promoters from Gram-positive bacteria for QS component expression could not function well in E. coli.

Then a engineered trc promoter (Ptrc) was used instead of native promoters for continuously active expression of QS components, as the lac operator sequence was deleted from trc promoter to abolish the Lac repressor binding for continuously active. It was found that in this time, each Gram-positive QS component-fusion was shown to express well and the prgX one-component QS system from E. faecalis (QEX) exhibited the highest dynamic range among all the testing systems, while the agr two-component QS system from S. aureus (QSR) presented the lowest dynamic range. The low response signal of QSR system was probably due to its QS component protein toxicity, as cell growth was impeded and final $OD_{600}$ decreased by 85% compared with the control strain (Fig. 1d).

**Crosstalk evaluation between different QS systems.** In contrast to previous pioneering studies exposing receptors to different concentrations of signaling compounds to investigate compatible sets of QS systems[12], we coupled ligand signal synthase and regulator-reporter system to investigate cross-interactions between different QS systems, as directly exposing to certain concentrations of ready-made ligand signal cannot monitor the dynamic behavior of accumulating ligands by synthase. Most importantly, some synthases may produce more bioactive ligand signaling species than we expected when overexpressed in E. coli[11]. With the six functional QS systems (QPB, QPA, QPH, QVX, QEX, QSR), all possible 84 response promoter/signal inducer/actuator combinations were created and the resulting GFP production was compared with the control strain harboring only response promoter and two corresponding empty plasmids. Only non-canonical promoter/inducer/actuator sets that exhibited more than 2-fold induction were defined as crosstalk.

As seen from Fig. 2a, although QS system from V. fischeri has been used in numerous circuits, a significant amount of crosstalk was observed between QS systems from V. fischeri and P. aeruginosa. The lux response promoter (PluxI) from V. fischeri could be activated by the expression of LasR and LasI or LuxR and LasI. LasR and LuxI in a certain extent could activate the LasB response promoter (PlasB) and LasA response promoter (PlasA), while HcnABC promoter (PhcnABC) is completely orthogonal to lux system.

The QS system from E. faecalis exhibited completely orthogonal to lux or las system from Gram-negative bacteria. In strains containing LuxR or LasR, expression of heptapeptide synthase CcfA could not activate the response promoter of PluxI, PlasB, PlasA or PhcnABC (Fig. 2a), indicating by no increase in fluorescence. On the other hand, when PrgX replaced LuxR or LasR, no increase in GFP expression was also observed using PluxI, PlasB, PlasA or PhcnABC as response promoter whether in the presence or absence of corresponding signal inducers (Fig. 2a). When GFP was placed downstream of response promoter from E. faecalis (PprgQ) in strains harboring LuxR (Fig. 2b), LasR (Fig. 2c), or AgrAC (Fig. 2d), constitutive fluorescence was observed with or without the CcfA expression. This suggested that the LuxR, LasR, and AgrAC proteins does not interact with the operator sequences in response promoter PprgQ for prgX, thus this response promoter is always active.

The QS system from S. aureus also exhibited orthogonal to lux or las system from Gram-negative bacteria. For instances, expression of LuxR and LasR or LuxI and LasI could not activate the response promoter from las or lux QS systems in strains containing AgrBD or AgrAC (Fig. 2a). However, the toxicity of its QS components was still observed in these hybrid QS systems. A dramatic decrease in the cell growth (final $OD_{600}$) of these strains, which contained part of QS components from S. aureus, was observed, further strengthening the hypothesis of QSR component toxicity (Supplementary Fig. 2).

**Rewiring QVX-controlled gene expression mechanism from positive feedback loop to on-time transcription circuit for decreasing leakiness.** As QS system from V. fischeri and E. faecalis exhibited high dynamic range and completely orthogonal characteristics, these two candidate QS systems were chosen for

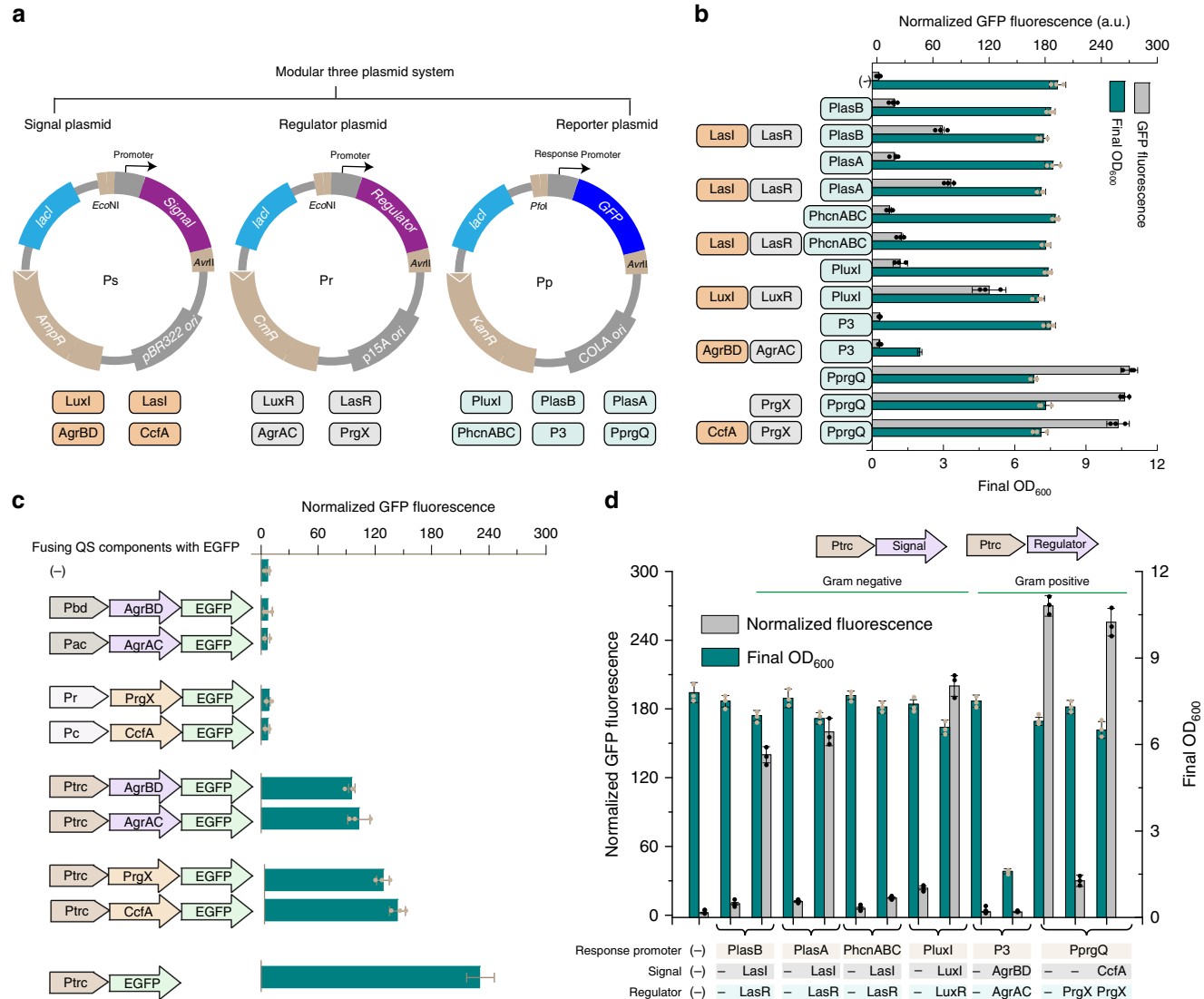

**Fig. 1 Characterization of QS systems from both Gram-negative and -positive bacterial species. a** Schematic of a modular three-plasmid system, including a signal plasmid (Ps), an regulator plasmid (Pr), and a reporter plasmid. **b** Validating the function of each QS system using native promoters for QS component expression. **c** Fusing QS components with GFP to measure translational output directly. Pbd, native promoter of AgrBD; Pac, native promoter of AgrAC; Pr, native promoter of PrgX; Pc, native promoter of CcfA; Ptrc, constitutive trc promoter. **d** Validating the function of each QS system using constitutive Ptrc promoters for QS component expression. The function of each QS system was validated by comparing normalized GFP fluorescence from strains harboring plasmids of Ps, Pa, and Pp to strains only harboring Pp. All fluorescence measurements were normalized by dividing measured fluorescence values by the $OD_{600}$ of that well to conduct a per-cell measurement. The recombinant strains were grown in 25 mL of MOPS medium at 30 °C with 220 rpm orbital shaking. Cell fluorescence and cell density ($OD600$) were measured after 30 h of culture on a Cytation 3 imaging reader system (BioTek, Winooski, USA). (−) indicated the wild type strain *E. coli* MG1655 with corresponding empty plasmids as the negative control. Values are shown as mean ± SD ($n = 3$ biological replicates). Source data are available in the Source data file.

further study. However, both of PluxI and PprgQ response promoters remained a residual activity in the absence of signaling molecules, which was defined as leakiness (Fig. 1d). This would impede the capability to investigate the effect of low expression levels or retain a gene in the off state, especially required for toxic proteins. As positive feedback loop, in which a low constitutive transcription of *luxI* leads to sufficient level of auto-inducer to further activate PluxI promoter, is the primary regulatory control over lux regulon, this makes leakiness of this promoter as a natural and un-avoiding trait (Fig. 3a).

The PluxI response promoter is located in ~200 bp regulatory region, which separated two operons of lux regulon. Although previous work demonstrated that a palindromic structure (a 20-nt lux box) upstream of *luxI* in this regulatory is vital for

positive-feedback response[27], we conceived whether there existed control region for initial constitutive expression and the regulatory mechanism of lux regulon could be rewired from positive-feedback loop to on time transcription by rational promoter engineering. To test this hypothesis, the leakiness level (Fig. 3b) and response behavior (Fig. 3c) of PluxI promoter was monitored by deletion mapping of regulatory region from both leftward and rightward operons extending into lux box.

The different variants of PluxI promoters were fused to GFP and the effect of these deletions on both promoter leakiness and response performance was evaluated by measuring output GFP fluorescence. When deleting from downstream side of the operon, the leakiness level of PluxI promoter decreased dramatically (I1-I3), which was measured in the absence of signal inducer and

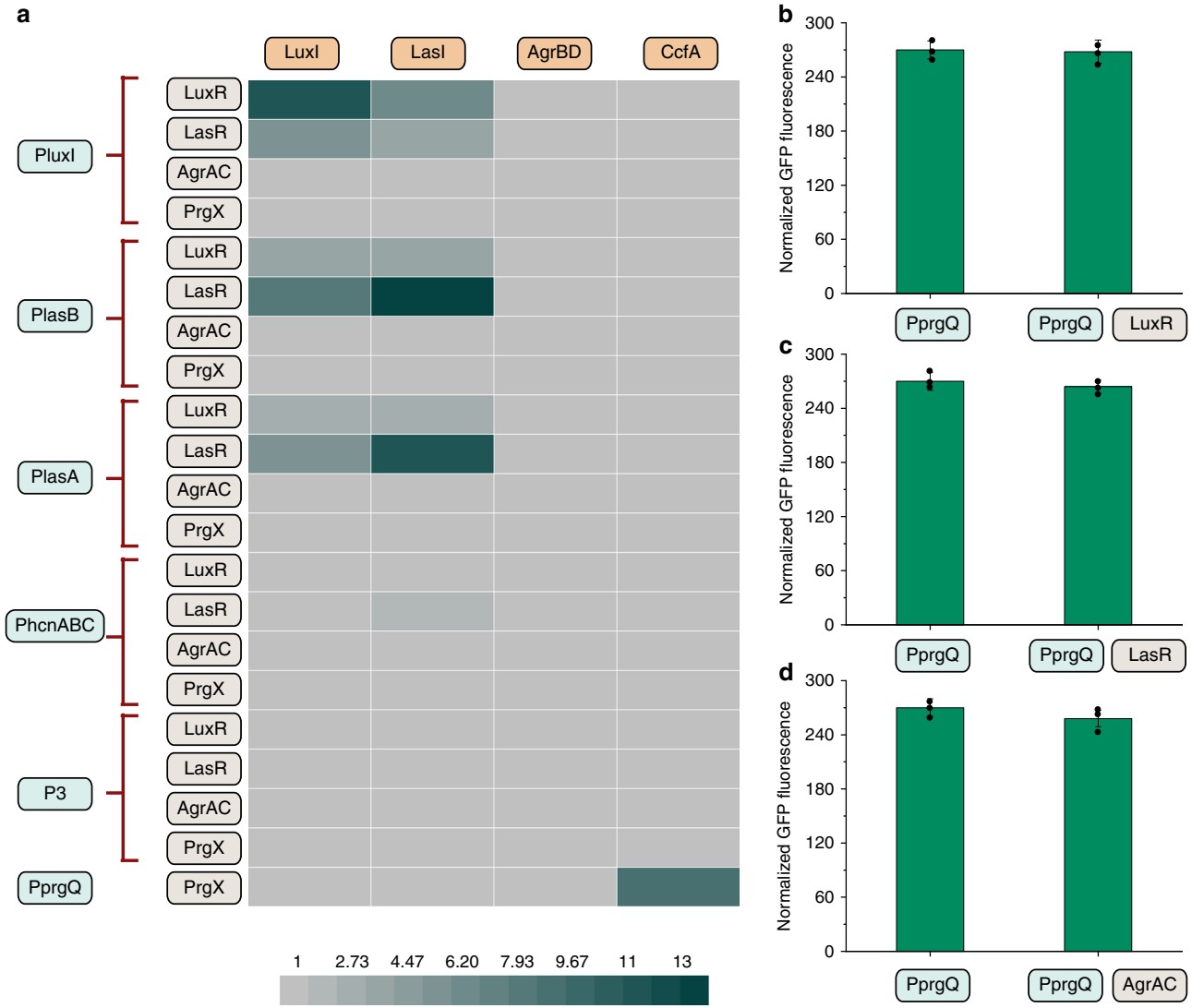

**Fig. 2 Investigating cross-interactions between different QS systems. a** Heat map of induction ratio from 84 combinations of different QS components. All possible 84 response promoter/signal inducer/actuator combinations were created. The induction ratio was defined as dividing the normalized GFP fluorescence from different combinations by the control strain harboring only response promoter and two corresponding empty plasmids. **b** Evaluating cross-talk between response promoter from *E. faecalis* (PprgQ) and regulator LuxR. **c** Evaluating cross-talk between response promoter from *E. faecalis* (PprgQ) and regulator LasR. **d** Evaluating cross-talk between response promoter from *E. faecalis* (PprgQ) and regulator AgrAC. Each QS component was expressed by constitutive trc promoter. Pre-cultured recombinant strains were grown in 25 mL of MOPS medium at 30 °C with 220 rpm orbital shaking. Cell fluorescence and cell density ($OD_{600}$) were measured after 30 h of culture on a Cytation 3 imaging reader system (BioTek, Winooski, USA). All fluorescence measurements were normalized by dividing measured fluorescence values by the $OD_{600}$ of that well to conduct a per-cell measurement. Values are shown as mean ± SD ($n = 3$ biological replicates). Source data are available in the Source data file.

actuator (Fig. 3b). However, the response function of these mutant PluxI promoters, which was measured in the presence of signal inducer and actuator, also showed a dramatic decrease (Fig. 3c), suggesting that this region was vital for promoter function. Whereas deleting from upstream side of the operon, the leakiness level remained unchanged until 69 bp from leftward operon (I4-I7), and commenced dramatically decreasing from 69 bp to 109 bp (I7-I9). Only a slight decrease of leakiness level was observed through deleting from 109 bp to 139 bp (I9-I11). Quantitative and statistical analysis of microscopy images of engineered strains harboring different lengths of response promoter fused to GFP were shown in Table S1 and S2, respectively. Notably, the response function of engineered PluxI promoter followed almost similar induction dynamics during deleting these regions (I4-I11 in Fig. 3c), thus deducing that regulatory sequences upstream of lux box (69–139 bp from leftward operon) are not necessary for auto-inducer and actuator dependent activation but are responsible for initial constitutive expression.

To further demonstrate this hypothesis, mutant PluxI promoters exhibiting dramatic leakiness decrease (I9-I11) were used to replace native PluxI promoter for driving auto-inducer expression in QVX genetic circuits (Fig. 3d). Whether we used native PluxI response promoter or mutant PluxI response promoter as output promoter, the lux system no longer responded to cell growth and the positive feedback loop could not be observed. The response function could only be presented when using native PluxI promoter for auto-inducer expression, demonstrating that positive feedback loop character of lux system has been rewired to low leakiness (9.5-fold decrease) on time transcription without hampering response function.

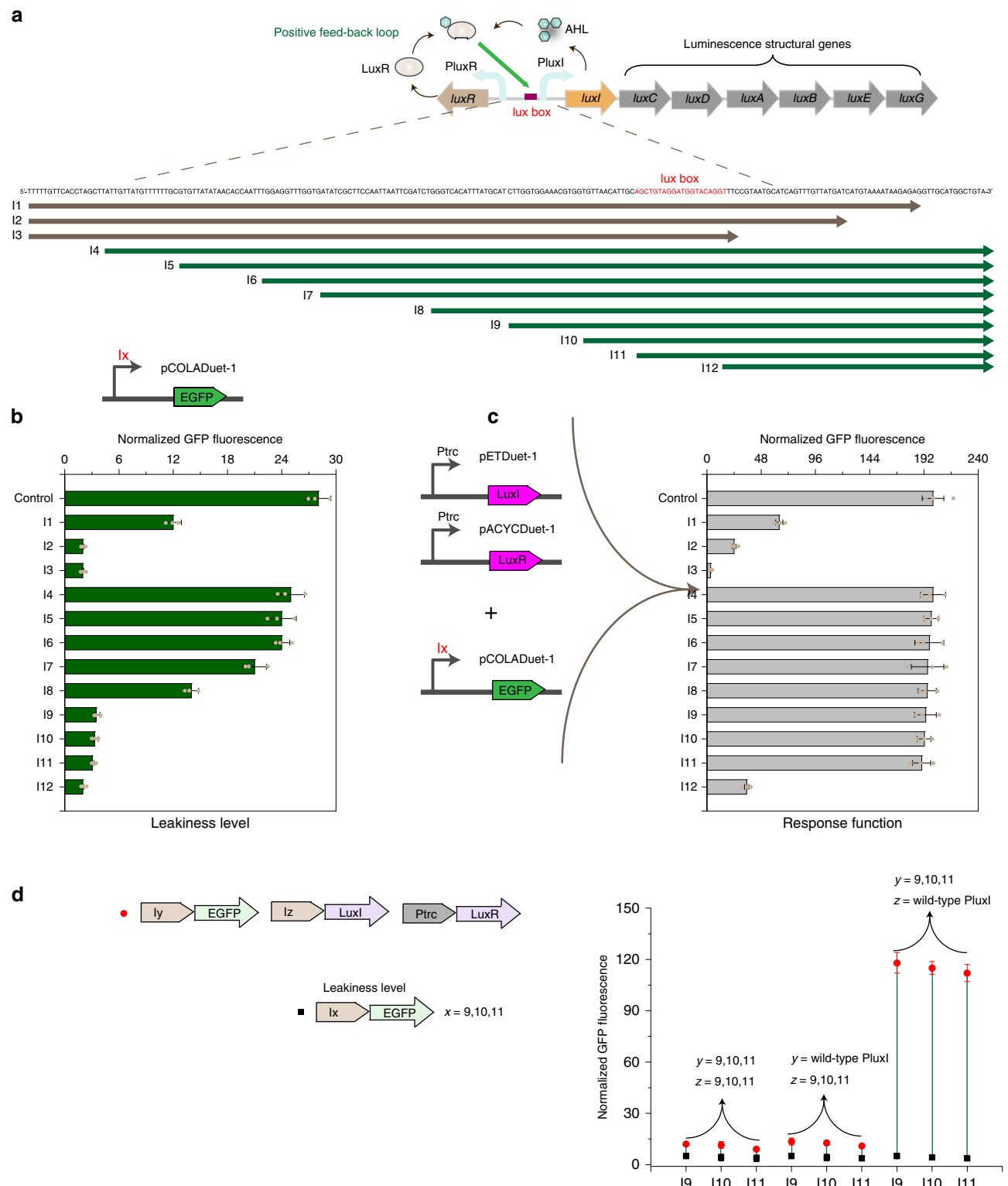

**Protein regulator engineering to decrease leakiness level of QEX system**. As QEX system also suffered from output leakiness (Fig. 1d), we increased the expression level of protein regulator PrgX, which is responsible for repressing prgQ operon, to prevent the undesirable leakiness (Fig. 4a). A set of six pre-characterized constitutive promoters (P1–P6: promoter strength ranking from high to low)[28] covering a large expression space were used to achieve a spectrum of regulator protein expression level (Fig. 4b). Based on our previous studies[29,30], the plasmid pCDFDuet-1

having higher gene copy numbers than initial pACYCDuet-1 was also employed to improve PrgX expression level. To verify its improved gene expression level, PrgX protein was firstly GFP tagged to measure its translational output using different promoters and plasmids. The chimera fluorescence matched well with the strength of promoters or plasmid gene copy number (Fig. 4b), demonstrating tunable expression level of PrgX.

It was found that increasing the expression level of PrgX would lead to a decline in QEX leakiness, which was demonstrated by

**Fig. 3 Rewiring QVX-controlled gene expression mechanism from positive feedback loop to on-time transcription circuit for decreasing leakiness.**
**a** Schematic of deletion mapping of the regulatory region between LuxR and LuxI. I1-I3 engineered promoters were deleted from downstream side of regulatory region between LuxR and LuxI, while I4-I12 from upstream side of regulatory region. **b** The leakiness level of engineered PluxI promoters with different deletions. Different variants of PluxI promoters were fused to GFP to monitor their leakiness levels using pCOLADuet-1. Control indicated wild-type PluxI promoter with the whole regulatory region between LuxR and LuxI. **c** The response function of engineered PluxI promoters with different deletions. QVX signal and regulator components were introduced into strains harboring different variants of PluxI promoters under Ptrc promoter using pETDuet-1 and pACYCDuet-1, respectively. Engineered strains were grown in 25 mL of MOPS medium at 30 °C with 220 rpm orbital shaking. Cell fluorescence and cell density ($OD_{600}$) were measured after 30 h of culture on a Cytation 3 imaging reader system (BioTek, Winooski, USA). All fluorescence measurements were normalized by dividing measured fluorescence values by the $OD_{600}$ of that well to conduct a per-cell measurement. **d** Microscopy images of engineered strains harboring different lengths of response promoter fused to GFP. Cells after 30 h of culture were photographed using a Nikon Eclipse 80i microscope (Nikon, Tokyo, Japan). Values are shown as mean ± SD ($n = 3$ biological replicates). Source data are available in the Source data file.

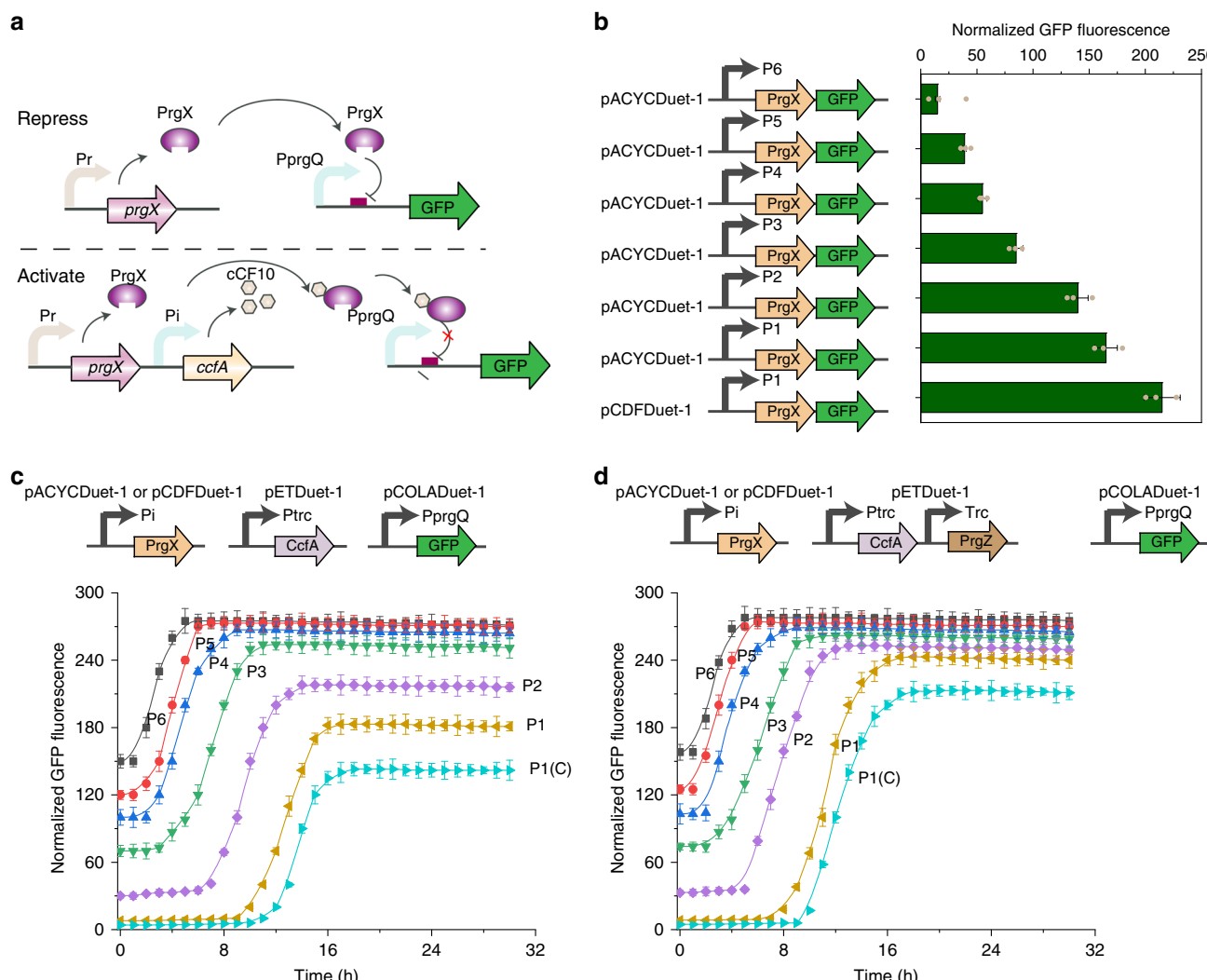

**Fig. 4 Decreasing leakiness level of QEX system by protein regulator engineering. a** Schematic of QEX-controlled gene expression mechanism. **b** The demonstration of tunable expression level of PrgX. PrgX protein was GFP tagged to measure its translational output using different promoters and plasmids. Engineered strains were grown in 25 mL of MOPS medium at 30 °C with 220 rpm orbital shaking. Cell fluorescence and cell density ($OD_{600}$) were measured after 30 h of culture on a Cytation 3 imaging reader system (BioTek, Winooski, USA). **c** The impact of different expression levels of PrgX on QEX leakiness level and response function through plate reader experiments. Response promoter PprgQ tagged with GFP was placed on pCOLADuet-1. CcfA was placed on pETDuet-1 under Ptrc constitutive promoter. P1, P2 indicated the expression of PrgX under P1, P2 promoter on pACYCDuet-1, while P1 (C) indicated expression of PrgX under P1 promoter on pCDFDuet-1. **d** The impact of introducing PrgZ on QEX response function through plate reader experiments. PrgZ was co-expressed with CcfA under Ptrc promoter on the same plasmid. Pre-cultured recombinant cells through overnight incubation at 37 °C were diluted to $OD_{600}$ of 0.01 in 96-well plate in 200 μL of MOPS medium. The culture was conducted on an Infinite M1000 PRO (Tecan, Switzerland) plate reader at 30 °C. Cell density and fluorescence were measured every 1 h for 30 h. Values are shown as mean ± SD ($n = 3$ biological replicates). Source data are available in the Source data file.

normalized GFP fluorescence of initial culture stage of QS circuits. However, high expression level of repressor would result in a delay of triggering times at the same expression of CcfA and influence its response function (Fig. 4c). As seen from Fig. 4c, strains with the highest expression level of PrgX (P1(C)) produced the lowest peak fluorescence and the latest triggering time was also observed. For avoiding these unfavorable traits, we sought to investigate whether increasing the sensitivity to signal peptide cCF10 by introducing a surface cCF10-binding protein PrgZ, which is known to mediate pheromone import[31–33], could improve circuit performance at the high expression level of PrgX (Fig. 4d). It was found that this offered an improvement in QEX performance, including dynamic range and triggering time. Of the seven different engineered circuits, engineered circuits P1 exhibited a good enough response such as low leakiness and high dynamic range.

**Design of a pathway-independent and full-autonomous global resource allocation strategy**. As shown in supplementary results (Supplementary Fig. 4), given the characters such as no-crosstalk, high dynamic range and low leakiness of QVX and QEX circuits, we sought to layer these two genetic switch modules associate with endoribonuclease MazF, which would cleave approximately 96% of *E. coli* coding RNA sequences[1], to explore whether intracellular resource allocation could be manipulated toward a required circuit behavior. A resource reallocation mechanism needs to be developed to preserve target processes while repressing competing pathways.

The impact of MazF expression on a target gene GFP was characterized in engineered *E. coli* strain with deletion of endogenous *mazF*, and QEX and QVX circuits were using to drive the expression of MazF and GFP, respectively. The native GFP containing 27 recognition sites of MazF in coding RNA sequence (GFP(N)) and the modified GFP using alternative codon to remove recognition sites (GFP(M)) were used in comparison. GFP(N) and GFP(M) driven by QVX circuit exhibited similar fluorescence, demonstrating that recording RNA sequences did not modify expression (Fig. 5a). Cellular resource redistribution activity was quantified by fold change defined as the ratio of normalized GFP fluorescence in the presence to absence of MazF after 30 h at 30 °C.

Following MazF induction, it was found that different triggering times of QEX circuits driving MazF expression imposed different impact on fluorescence dynamics (Fig. 5b). Suitable triggering time of QEX P3 MazF circuit acting on QVX P2 GFP circuit presented the best performance (~3-fold increase in fold change), while an early or delay triggering time points would lead to a decrease in fold change (Fig. 5c). The early triggering of MazF might dramatically shorten normal growth period, thus limiting cellular resources for minimum cell growth, as the final $OD_{600}$ decreased dramatically by 87.1% (Supplementary Fig. 6). Whereas the delay triggering cannot efficiently prevent cellular resources from cell growth toward target functions. The fold change for GFP-N was always below 1 under MazF expression driven by each QEX circuit (Supplementary Fig. 7), further demonstrating the capability of MazF for enhancing protected and inhibiting unprotected gene expression.

As auto-inducer synthase and transcriptional regulator of QS systems are enriched for MazF recognition sites (Gene sequences shown in Supplementary materials), these four proteins (LuxI, LuxR, and CcfA, PrgX) were recoded to remove these sites. Native QS system components of QEX P3 MazF and QVX P2 GFP circuits, which exhibited maximum fold change (Fig. 5c), were replaced by these modified candidates (LuxI(M), LuxR(M), CcfA (M), PrgX(M), PrgZ(M)) to investigate whether this could

improve circuit performance (Fig. 5d). Cells bearing LuxI(M) and LuxR(M) exhibited a higher GFP expression compared to cells expressing CcfA(M), PrgX(M), and PrgZ(M) (6.8 VS 3.5-fold change), and a combination of these modified proteins yield the best circuit performance (approximately 11-fold change).

Considering the positive effect of using modified QEX components, which was directly related to MazF abundance, we investigated whether protection of MazF (Supplementary Fig. 8) or deleting MazF inhibitor (MazE) (Supplementary Fig. 9) would also enhance resource redistribution activity. Contrary to expectation, although expression of MazF(M) exhibiting a higher MazF mRNA level than cells expressing native MazF, lower values of fold change was observed (Supplementary Fig. 8). This demonstrated the importance of establishing a MazF mRNA-decay negative feedback loop, which might avoid the MazF overexpression related effect[1]. Moreover, deletion of MazE also exerted no observable effect on the circuit performance (Supplementary Fig. 9), demonstrating the enough MazF abundance in this circuit.

**Using resource allocator circuits to enhancing metabolic production yields**. As shown in supplementary results (Supplementary Fig. 10), we also sought to examine essential factors that could improve the resource allocator performance after protection from MazF. However, an exciting application of the resource allocator would be to use it as a tool for optimizing biotechnological processes. Commercially interesting compound medium chain fatty acids (MCFAs), which constitute important components of fuels, commodities and fine chemicals[34], were selected. The engineered reversal of the β-oxidation cycle (r-BOX), consisting of *Ralstonia eutropha* thiolase (*bktB*), *E. coli* 3-hydroxyacyl-CoA dehydrogenase/enoyl-CoA hydratase (*fadB*), *Euglena gracilis* trans-enoyl-CoA reductase (*ter*), and *E. coli* thioesterase (*ydiI*) (Fig. 6a) for MCFA biosynthesis, was driven by QVX circuit with different triggering times, and each gene was placed under pLuxI I11 response promoter control (Fig. 6b). It was found that QVX circuit with P2 driving QS component expression achieved higher MCFA titer (180.6 mg/L) than other candidates (Fig. 6b, top panel). However, this titer is still lower than our previous engineered strains (313 mg/L)[26] using inducible promoters with the same set of pathway enzymes.

This lower MCFA titer outcome led us to suspect that dynamic range of QVX circuit may be inadequate for MCFA biosynthesis. Hence, higher copy number of pETDuet-1 plasmid used in our previous studies[30,34] was employed as response plasmid for driving MCFA biosynthetic pathway. As seen from Fig. 6b, middle panel, increasing gene copy number of plasmids improved MCFA titer to 325 mg/L when using pETDuet-1. Quantitative real time PCR was performed to confirm pathway enzyme expression levels (Supplementary Fig. 12). We also examined the effect of using native leaky PluxI responsive promoter for driving MCFA biosynthetic pathway on MCFA titers (Fig. 6b3), and it was found that the early imposing metabolic burden on cells due to its leaky character led to lower MCFA titers and final $OD_{600}$. This further demonstrated the importance of decreasing QVX system leakiness.

In the next step, QEX circuits with different triggering times were used to drive MazF expression in strains harboring QVX P2 MCFA circuit (Fig. 6c), which exhibited the highest MCFA titer among the above-mentioned candidates. Both unprotected and protected MCFA pathway enzymes were overexpressed, respectively. Activity profiles of resource redistribution for these strain series, which were defined as fold change of MCFA titers in the presence to absence of MazF, trended similarly with previous characterization of GFP library. An early or delay of MazF

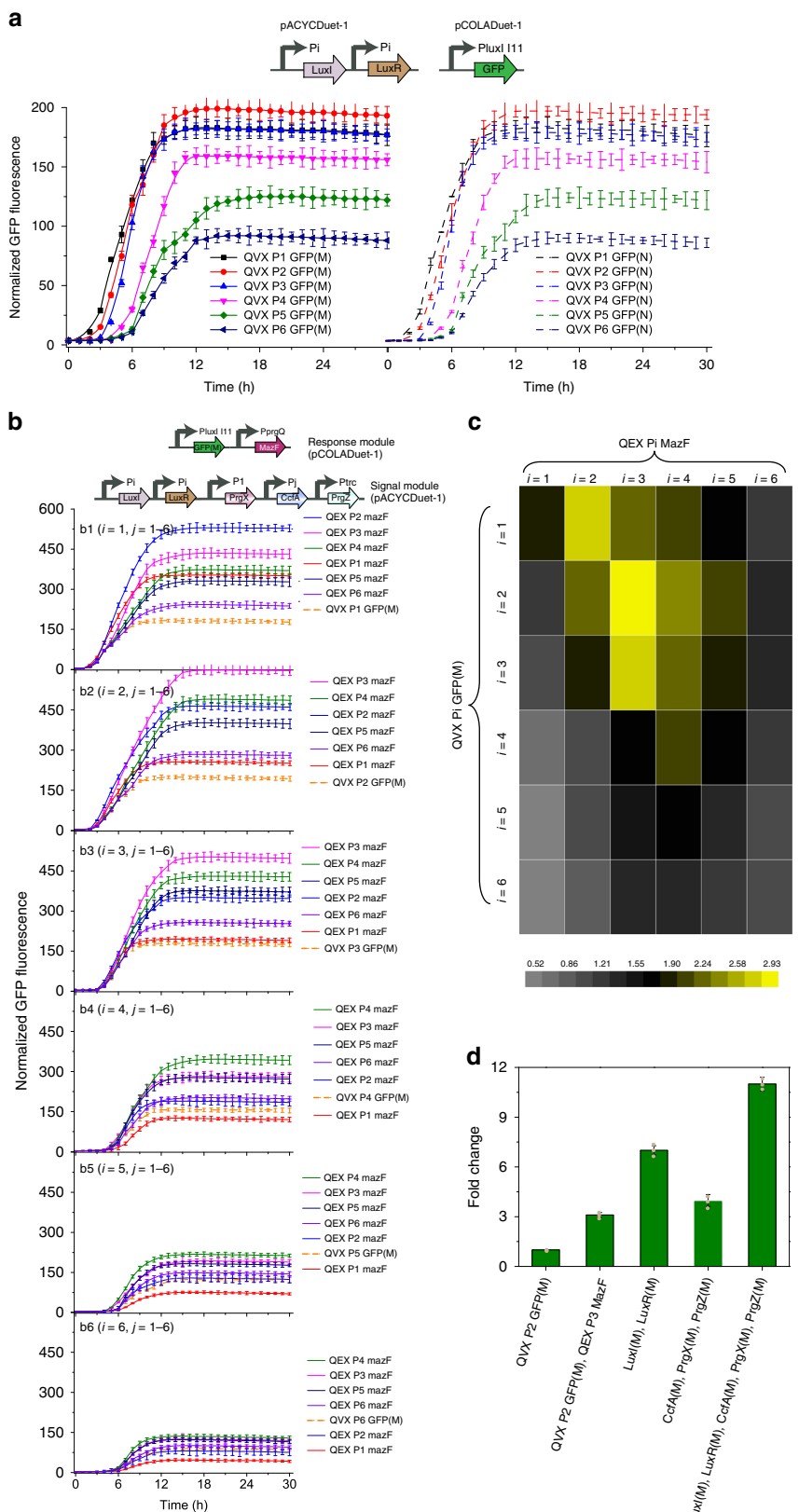

triggering time would incur suboptimal redistribution activity and a boost in MCFA titer was observed with an intermediate triggering time points (Fig. 6c). Engineered strain with QEX P3 MazF circuit exhibited 97% (Fig. 6c, top panel), 491% (Fig. 6c, middle panel) increase in total MCFA titer and specific MCFA titers, respectively, compared to control strains without MazF

expression, demonstrating the capability of this resource allocator circuit to redirect resources demanding for building blocks toward a target product biotechnological processes. Furthermore, total MCFA titers of engineered strains with unprotected pathway enzymes were lower than control strains without MazF expression (Fig. 6c, top panel), further demonstrating the capability of

**Fig. 5 Design of a pathway-independent and full-autonomous global resource allocation strategy. a** Comparing normalized fluorescence of GFP-N and GFP-M driven by QVX systems. PluxI I11 indicated engineered PluxI promoter of I11. QVX Pi GFP(M) or QVX Pi GFP(N) indicated normalized fluorescence of GFP(M) or GFP(N) driven by QVX system using Pi promoter for LuxR and LuxI expression (i = 1–6). **b** The impact of different triggering times of MazF expression on fluorescence dynamics driven by QVX system using P1–P6 promoters (B1–B6) for LuxR and LuxI expression. QEX Pi MazF indicated introducing MazF driven by QEX circuit using Pi promoter for CcfA expression into control strains harboring QVX Pi GFP(M) circuits (i = 1–6). **c** The impact of MazF expression on fluorescence fold change. **d** The effect of protecting QS system components from MazF on fluorescence fold change. QVX P2 GFP(M) indicated GFP(N) expression driven by QVX system using P2 promoter for LuxR and LuxI expression. QVX P2 GFP(M), QEX P3 MazF indicated introducing QEX P3 MazF circuit into strains harboring QVX P2 GFP(M) circuit. LuxI(M), LuxR(M) indicated replacing LuxI and LuxR with LuxI(M) and LuxR(M) in strains harboring QVX P2 GFP(M), QEX P3 MazF. CcfA(M), PrgX(M), PrgZ(M) indicated replacing CcfA, PrgX, and PrgZ with CcfA(M), PrgX (M), and PrgZ(M) in strains harboring QVX P2 GFP(M), QEX P3 MazF. LuxI(M), LuxR(M), CcfA(M), PrgX(M), PrgZ(M) indicated replacing LuxI, LuxR, CcfA, PrgX, and PrgZ with LuxI(M), LuxR(M), CcfA(M), PrgX(M), PrgZ(M) in strains harboring QVX P2 GFP(M), QEX P3 MazF. Pre-cultured recombinant cells were diluted to OD$_{600}$ of 0.01 in 96-well plate in 200 μL of MOPS medium on an Infinite M1000 PRO (Tecan, Switzerland) plate reader at 30 °C. Cell density and fluorescence were measured every 1 h for 30 h. Values are shown as mean ± SD ($n$ = 3 biological replicates). Source data are available in the Source data file.

this resource allocator for enhancing protected and inhibiting unprotected gene expression.

Activity profiles of resource redistribution again showed that protection of host factors would enhance MCFA titers (Fig. 6d), as characterized in initial GFP library. Protection of elongation factor EF-Ts, ribosomal protein subunits S9, S20, L17, and RNA polymerase subunits β, β′ from MazF activity led to an improvement of 64.1, 39.2, 32.8, 43.7, 46.9, and 39.1%, respectively, over control strain without host factor protection on MCFA titers. A combination of EF-Ts-M, β-M, and β′–M achieved a 1.5- and 1.6-fold increase of total MCFA and specific MCFA titers, respectively, indicating that discovering support factors in need of protection would augment resource redistribution activity. Notably, this final engineered strain exhibited a 4.1- and 14.2-fold increase of total MCFA and specific MCFA titers, respectively, over control strains without MazF expression. Furthermore, it was noticeable that suitable triggering time of MazF expression varied in cases of reporter gene fluorescence and MCFA production, further demonstrating the importance of independently controlling two different cellular processes and conferring a tunable advantage to our system.

**Evaluating the resource allocator circuit performance in bioreactors.** Despite the promising results from shake flask cultures, it is important to evaluate this resource allocator circuit in more industrially relevant procedures, such as in scaled-up bioreactors. Based on previous flask culture results, the resource allocator performance (Supplementary Fig. 13A) was evaluated in a 5-L fermenter, and it was found that resource allocator performance was even better than initial shake flask results (Supplementary Fig. 13B). With the implementation of dissolved oxygen (DO) (30%), glucose (5 g/L) and pH control (6.5), strain S_B3 produced a nearly 5.4- and 30.2-fold increase in MCFA titers and yields over control strain (S_B1) (Supplementary Fig. 13D). In addition, at the end of each fermentation (60 h), the concentrations of four main byproducts such as acetate, ethanol, lactate, and succinate were measured to investigate the characteristics of dynamic regulation metabolism. It was found that these byproduct concentrations decreased dramatically when implementing the resource allocators (Supplementary Fig. 13C). The better performance in bioreactors may be attributed to the character that minimizing unintended cell proliferation related impact in a complex environment, suggesting its great potential in scale-up commercial applications.

## Discussion

Developing strategies to switch between modes of growth and production for controlling resource economy of cells is a fundamental goal of biological chemical production[1]. Cellular resources and energy would be redistributed to specific target pathways when cells switch to production phenotypes, minimizing resource expenditure toward unessential cellular activities. QS-based dynamic pathway regulations provide a promising alternative as broadly applicable circuits to realize this phenotypic shift. However, current strategies only allow for a single QS circuit fully functional in one cell, and mainly provide localized control of cellular activities, lacking the capability to control two individual cellular states and redistributing global resource allocation[3].

Here, a dynamic, pathway-independent and full-autonomous global resource allocation strategy was constructed to achieve a switch between growth and production phenotype. The phenotypic shift including target pathway expression and global competing pathway inhibition was realized in a full-autonomous way, and eliminate human supervision totally during the overall fermentations. The implementation of this strategy in *E. coli* led to a 23.6-fold increase in reporter gene fluorescence (Fig. 5), and its application in biotechnological processes was further extending to MCFA production in both shake flask culture and 5-L bioreactors. This resulted in a 14.2-fold increase in specific MCFA titer in shake flask cultures (Fig. 6) and a 24.9- and 30.2-fold increase in specific MCFA titer and yield in bioreactors (Supplementary Fig. 13), demonstrating the broad utility and large-scale application.

Although previous pioneering work repurposed QS system for dynamical gene expression control in academic metabolic engineering projects[11–13], in most cases, no more than one type of QS system could be fully functional in a single cell. Previous studies mainly focused on characterizing lux-like QS systems from Gram-negative bacterium to investigate orthogonal pairs[11,12], often led to off-target binding or functional cross-talk. Another exciting recent work described up- and down-regulate expression of 2 sets of genes by combining components of lux and esa QS systems, still lacking the ability to simultaneously up-regulate two different sets of genes[18]. Here, to expand the screen range, numerous QS systems from both Gram-negative and -positive species based on ligand uniqueness were evaluated to identify orthogonal QS systems functional in a single cell. It was found that QS system from *Enterococcus faecalis* (QES) exhibited completely no cross talk with currently most widely used AHL-based circuits, and the highest dynamic range among all the testing systems (Fig. 1). These findings broaden the toolbox of characterized components for synthetic regulatory circuits.

Lux regulon from *Vibrio fischeri* is a well-studied example and analogous AHL-based circuits are one of the most favorite alternatives considered to design genetic devices[10,11]. However, as positive feedback loop is the primary regulatory control over lux

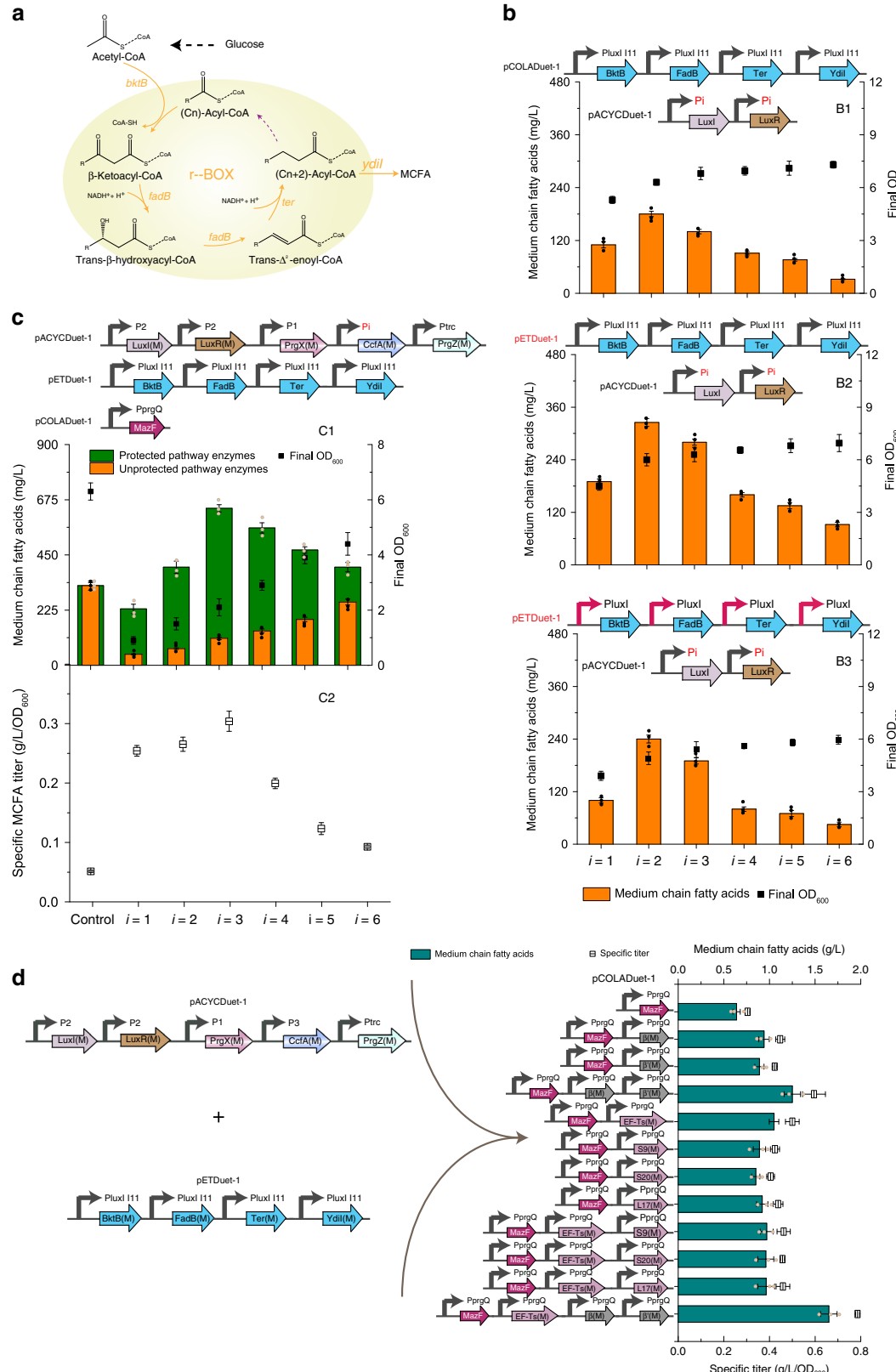

**Fig. 6 Enhancing metabolic production yields by resource allocator circuits. a** Schematic of r-BOX cycle. **b** The r-BOX cycle was driven by different triggering times of QVX circuits. Pi (i = 1–6) indicated using Pi (i = 1–6) promoters for LuxI and LuxR expression. (B1) PluxI I11 response promoters fused with each pathway enzyme were placed on pCOLADuet-1. (B2) PluxI I11 response promoters fused with each pathway enzyme were placed on pETDuet-1. (B3) PluxI I11 was replaced with native PluxI promoter. **c** The effect of different triggering times of QEX circuits driving MazF expression on MCFA production. Pi (i = 1–6) indicated using Pi (i = 1–6) promoters for CcfA expression. **d** The effect of protecting host factors from MazF on MCFA titers. Pre-cultured recombinant strains were grown in 25 mL of MOPS medium at 30 °C with 220 rpm orbital shaking. MCFA titers and cell density (OD$_{600}$) were measured after 48 h of culture. Values are shown as mean ± SD (n = 3 biological replicates). Source data are available in the Source data file.

regulon (Fig. 3a), this makes leakiness of the response promoter PluxI as a natural and un-avoiding trait, hindering the ability to examine the effect of low expression levels or retain a toxic protein in the off state. Here, deletion mapping of regulatory region between *luxR* and *luxI* was used to monitor the leakiness level and response behavior. For the first time, we found that regulatory sequences upstream of lux box are not necessary for second activation but are responsible for initial constitutive expression, and truncation of PluxI promoter would rewire positive feedback loop character to low leakiness on time transcription without hindering response behavior (Fig. 3). This new character renders AHL-based circuits to higher-level genetic circuitry for more broad applications such as exploring the effect of low expression level or toxic proteins, in synthetic biology.

Overexpressing heterologous enzymes or rate-limiting enzymes in natural pathways would quickly approach their limits because of the rigidity or robustness of the underlying regulatory networks, which evolve to counteract physiological or genetic perturbations and maintain flux distributions towards cell growth[35,36]. Instead of interrupting specific pathway activities, a fruitful choice would be to operate on global regulatory mechanisms directly, drawing away cellular resources and energy toward target metabolite production. Utilizing nutrient starvation is a well-demonstrated strategy, but at the expense of hampering production capacity of the cell especially when target compounds themselves consist of corresponding nutrient atoms[35].

Here, it was showed that coupling programmable mRNA decay to dynamic regulation mechanism, cellular resources could be reallocating into target pathways with the protection of engineered network and host factors, led to a 14.2- and 24.9-fold increase in specific MCFA titer in shake flask cultures (Fig. 6) and bioreactors (Supplementary Fig. 13). The better performance in fermenters might be due to the characteristic that minimizing unintended cell proliferation related impact and allowing cells executing desired function in complex environment, suggesting its great potential in scale-up commercial applications. In this study, it was demonstrated that discovering support factors in need of protection would augment resource redistribution activity (Supplementary Fig. 10). However, the impact of cell growth repression on cellular sub-systems involved in transcription, translation and replication, still remain elusive, and a thorough exploitation of cellular activities and resource allocation in engineered cells would further improve the behavior of target circuits.

## Methods

**Strains, plasmids, and general procedures.** *E. coli* strain MG1655 was utilized for all molecular experiments and bio-catalysis. The compatible plasmids including pCOLADuet-1, pACYCDuet-1, pCDFDuet-1, and pETDuet-1 (Novagen, Germany) should be existed in the same cell coupling with the addition of 40 μg/mL of kanamycin, 20 μg/mL of chloramphenicol, 40 μg/mL of streptomycin, 100 μg/mL of ampicillin, respectively. Genetic manipulation related reagents such as T4 DNA ligase, phusion DNA polymerase, and FastDigest restriction enzymes are purchased from Novagen (Darmstadt, Germany) and performed based on standard molecular biology. The detail information about the construction of QS systems, different variants of PluxI promoters, global resource allocators, and protection of host factors was shown in Supplementary methods in Supplementary Materials. All the primers used in this manuscript were provided in Supplementary Tables 1–9.

**Gene deletions.** Primers (Sangon Biotech, Shanghai, China) used in this section were shown in Supplementary Table 9. *mazF* and *mazE* were deleted from *E. coli* MG1655 utilizing a lambda-red recombination-based method[37]. Briefly, primers Pf_mazF-Kan$^{FRT}$ and Pr_mazF-Kan$^{FRT}$ were used to amplify *Kan$^{FRT}$*, and both primers incorporated 80 bp of homology with the ends of *mazF* gene to facilitate integrating into proper locus. Colony PCR and sequencing were utilized to verify proper colonies after transforming this cassette into *E. coli*. FLP recombinase-expressing pCP20 was then transformed into the resulting strains to mediate the excision of FRT-flanked Kan. Similarly, primers Pf_mazE-Kan$^{FRT}$ and Pr_mazE-Kan$^{FRT}$ were used to delete *mazE* from *E. coli* genome.

**Real-time PCR measurements.** Primers used in this section were shown in Supplementary Table 9. The transcriptional levels of target genes were confirmed by real-time PCR measurements and RNAprep Pure Kit (TIANGEN, Beijing, China) was used to prepare total RNA from engineered cells, which were collected at the stationary phase and immediately frozen in liquid nitrogen. According to our previous procedures[34], the conduct of real-time PCR was performed on Light-Cycler 480 II thermal cycler system (Roche, Mannheim, Germany) through SYBR green PCR Master Mix (TaKaRa, Dalian, China). Primers Pf_qα2(M)/Pr_qα2(M), Pf_qβ(M)/Pr_qβ(M), Pf_qβ‘(M)/Pr_qβ‘(M), Pf_qEF-Ts(M)/Pr_qEF-Ts(M), Pf_qS9(M)/Pr_qS9(M), Pf_qS20(M)/Pr_qS20(M), Pf_qL17(M)/Pr_qL17(M), Pf_qbktB/Pr_qbktB, Pf_qfadB/Pr_qfadB, Pf_qter/Pr_qter, Pf_qydiI/Pr_qydiI, which were designed by Primer 3 software (http://bioinfo.ut.ee/primer3-0.4.0/), were used to amplify α2, β, β′, EF-Ts, S9, S20, L17, *bktB*, *fadB*, *ter*, *ydiI*. The reference gene *cysG*, which did not change significantly in response to MazF activity based on previous RNA-seq experiments[1], was used to normalize qPCR values with primers Pf_cysG/Pr_cysG for amplifying. Three biological replicates were performed and averaged for each sample.

**Fluorescence intensity measurement.** For shake flask culture, the recombinant strains were firstly grown in LB medium plus corresponding antibiotics at 37 °C with 220 rpm orbital shaking for approximately 6–8 h, and then diluted to OD600 of 0.01 in 25 mL of MOPS medium. The MOPS medium was prepared according to previous study[38] supplemented with 10 g/L D-glucose. The culture was conducted at 30 °C and 220 rpm of orbital shaking, and cell fluorescence and cell density (OD$_{600}$) were measured after 30 h of culture on a Cytation 3 imaging reader system (BioTek, Winooski, USA). All fluorescence measurements were normalized by dividing measured fluorescence values by the OD$_{600}$ of that well to conduct a per-cell measurement. The wild type strain *E. coli* MG1655 was used as the negative control. The emission and excitation wavelengths are 520, 488 nm for GFP and 610, 584 nm for mKate2.

Time-course measurements were conducted by plate reader experiments. Recombinant cells were firstly grown in LB medium plus corresponding antibiotics at 37 °C with 220 rpm orbital shaking for approximately 6–8 h, and then diluted to OD$_{600}$ of 0.01 in 96-well plate in 200 μL of MOPS medium. The culture was conducted on an Infinite M1000 PRO (Tecan, Switzerland) plate reader at 30 °C. Cell density and fluorescence were measured every 1 h for 30 h. The emission and excitation wavelength were 510, 485 nm for GFP and 610, 587 nm for mKate2. For plate reader experiments, each experiment was performed in triplicate through inoculating into different wells in 96-well plates and the averaged values were used for analysis.

**Analytical methods.** The measurements of ethanol, acetate, succinate, and lactate were performed through high-performance liquid chromatography (HPLC) equipped with Shimadzu Prominence SIL 20 system (Shimadzu Scientific Instruments) and an HPX-87H organic acid column (Bio-Rad). The conducting conditions were performed as 30 mM $H_2SO_4$ mobile phase, 0.3 ml/min flow rate, column temperature 42 °C. For each measurement, triplicate cultures are conducted and an error bar was utilized to represent their deviation.

For free fatty acid quantification, HPLC and gas chromatograph mass spectrometry (GC-MS) were used to analyze both extracellular and intracellular fatty acids. Free fatty acids were extracted based on our previous study[34]. HPLC measurement of fatty acid phenacyl esters was performed according to previously modified protocol[39] along with Agilent 1100 series HPLC instrument and a reverse-phase Gemini NX-C18 column (5 × 110 mm). 300 μL of 10 mg/mL triethylamine and 300 μL of 10 mg/mL phenacyl bromide were mixed with MCFAs, and the mixtures were then placed in 65 °C water bath for 6 h to convert fatty acids to corresponding phenacyl esters. The operation conditions were set as below: 30% acetonitrile for 5 min, 60% acetonitrile for 10 min, 95% acetonitrile for 15 min, 30% acetonitrile for 5 min. GCMS-QP2010 Plus (Shimadzu) and Rtx-5 MS capillary column (length of 30 m, film thickness of 0.25 μm, and diameter of 0.25 mm) were utilized to conduct GC-MS analysis of MCFAs based on our previous study[34]. Three biological replicates were performed and averaged for each sample.

Agilent 1260 HPLC system along with ZORBAX Carbohydrate column (5 μm, 4.6 × 150 mm) and a refractive index detector (Agilent RID 1260) were used to measure residual D-glucose. The conduction conditions were set as previous protocols[39]: 30 °C for keeping columns, 50% ACN and 50% water for mobile phase, a flow rate of 1 ml/min. Each measurement was performed in triplicate replicates, and an error bar represented as standard deviation (SD) with 95% confidence interval (CI) was utilzed to show the deviation.

**Culture conditions.** For microbial MCFA fermentations in shake flask cultures, the pre-inocula, which was prepared through 6–8 h of incubation in LB medium plus corresponding antibiotics, was diluted to OD$_{600}$ of 0.01 into 50 mL MOPS medium[38] supplemented with 10 g/L D-glucose. The culture was conducted at 30 °C and 220 rpm. MCFA titers and cell density (OD$_{600}$) were measured after 48 h of culture.

**Batch culture.** Seed culture was grown on rotary shakers for 6–8 h (37 °C, 220 rpm) an then diluted into 3-L BioFlo 115 fermentor (New Brunswick Scientific

Co, Edison, NJ), which contained 1.5 L MOPS minimal medium plus corresponding antibiotics and 10 g/L D-glucose. The D-glucose concentration was maintaining at 5 g/L during fermentation by supplementing with concentrated D-glucose (800 g/L). The cultivation temperature was maintained at 30 °C. The pH was kept at 6.5 by automatically feeding 12.5% NH₄OH solution or phosphoric acid solution. The dissolved oxygen concentration was set at 30% saturation through agitation cascade (200–500 rpm). MCFA fermentations were performed in triplicate, and an error bar represented as SD with 95% CI was used to show the deviation.

**Reporting summary**. Further information on research design is available in the Nature Research Reporting Summary linked to this article.

## Data availability
Data supporting the findings of this work are available in the manuscript and Supplementary Materials. All the datasets generated and analyzed in the paper are available from corresponding author upon request. All the sequences of engineered genes are provided in Supplementary sequences in Supplementary Materials. Any other relevant data are available upon reasonable request. Source data are provided with this paper.

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

## Acknowledgements
This work was financially supported by the Fundamental Research Funds for the Central Universities (KYGD202003), National Natural Science Foundation of China (No. 31972060), China Postdoctoral Science Foundation (2018M640491), Postdoctoral Research Funding of Jiangsu Province (2018K030B), and the Priority Academic Program Development of Jiangsu Higher Education Institutions (PAPD).

## Author contributions
J.J.W. designed the experiments, performed the plasmid construction and MCFA production, and wrote the paper. M.J.B. performed the protein fluorescence experiment. X.G.D., C.W.C., J.H.G., and S.Y.C. performed the HPLC and GC-MS related experiment. Q.Q.Z. and Z.J.Z. performed the fermenter experiment.

## Competing interests
The authors declare no competing interests.

**Additional information**

