## [Peer Review File · Nature Communications]

Reviewers' Comments:

Reviewer #1:

Remarks to the Author:

"Developing a pathway-independent and full-autonomous global resource allocation strategy to dynamically switch phenotypic states" by Wu and colleagues describes the use of quorum sensing circuits to change the resource allocation between host and synthetic circuits during regulation. They first demonstrate this with a fluorescent reporter, but also show a metabolic engineering application of the production of fatty acids. The system described is a moderate advance on the work of the Prather lab where a quorum sensing system was used to direct flux from glycolysis for the production of myo-inositol and glucaric acid (Gupta et al, 2017, Nature Biotech, 35:273-279) in that it both redirects flux and also induces the expression of the synthetic circuit by using two orthogonal quorum regulators. It is also a minor advance on the work of Adam Arkin's group who already showed that MazF could be used for global resource allocation in *E. coli* when expressed (Venturelli et al, 2017, Nature Comms, 8, Article number: 15128). In the present case, MazF is autoinduced by internally synthesized AHL and the effect of recoding key host genes needed for expression (RNA polymerase, TFs, etc) is investigated.

My major criticism of this paper is its sheer size. At 126 pages, the submitted version is almost the length of a PhD thesis. There are 11 figures and 8 tables in the main manuscript, which is well above the number allowed by the journal. The tables, especially, do not add anything to the main text and could have been part of the supplementary information. I would have liked to see the authors take the time to craft a clear, concise message from the data before submitting it for review. I understand that a lot of time and effort went into doing these experiments, but the temptation to show it all in the main text should be avoided. More is not always better.

The English could use minor editing in places for subject-verb agreement, correct use of adjectives and adverbs, etc.

Specific comments:

-Page 7, lines 123-125, The authors say, "Although there are a few pioneering studies characterizing different QS systems to investigate orthogonal pairs (ref 11-13), no more than one type of QS system could be co-expressed in a single cell." The reference list here is incomplete as there have been more than a few studies in this area over the last few years. In addition, it is not clear what the authors mean by, "no more than one type of QS system could be co-expressed". This is phrased as to mean it is not possible, but maybe the authors mean it has not been attempted?

-Page 10, line 171, referencing three of their own papers to discuss a three plasmid system that has three different origins of replication and three different selection markers seems gratuitous.

-Page 10, lines 176-8, The authors say they compared the strains with three plasmids to a control that just had the reporter plasmid. This is not a properly controlled experiment because of the effects of maintaining multiple plasmids and synthesis of three antibiotic resistance markers on the cells. These studies should have been conducted using the reporter plasmid and two empty plasmid backbones corresponding to the Ps and Pa plasmids. While normalizing GFP fluorescence by the OD600 can control for small differences in growth rates, the fluorescence/OD is not perfectly linear due to the effects of light scattering on fluorescence measurements.

-Bar charts should show individual data points as well as the mean and standard deviation

-Figure 2D, where are there three bars for the PprgQ system? Is this to show that it is constitutive in the absence of its regulator?

-Lines 267 ff, 'leftward' and 'rightward' operon don't seem particularly technical terms.

-Why are the mKate samples in Figure 7 labelled as 'EFP'?

-I think you mean 'recoded' where you have said 'recorded'

-Is the data in Figure 8B representative curves (n=1) or were replicate measurements taken? It would be helpful to include error bars to understand the reproducibility of various autoinduction conditions as well as whether individual conditions are different from each other

-Page 25, line 480. I would suggest changing the title of this section. You have not really investigated the stability of the circuit, but rather its performance under industrial conditions. I also do not understand the reason why overflow metabolites are presented as part of this study.

Reviewer #2:

Remarks to the Author:

The authors tested four quorum-sensing systems from different bacteria and found that the quorum-sensing systems from *Vibrio fischeri* and *Enterococcus faecalis* can be further engineered to achieve tight and orthogonal control of gene expression in *E. coli*. The timing of the gene expression of the quorum systems can be further fine-tuned by varying the expression of regulators, which allows differential gene expression during cellular growth. They used the endoribonuclease *mazF* to divert cellular resources to the output gene expression and enhance GFP or target molecules, by protecting genes from *mazF*-mediated RNA degradation. The output expression was significantly enhanced when both the *mazF* and output genes are expressed from the quorum-sensing systems, with specific fine-tuned timing. To further increase the output gene expression, the authors protected the regulators of the quorum-sensing circuit and also the transcriptional and translational machinery. Using the same system, they introduced four metabolic genes for the reverse beta oxidation pathway and demonstrated an increase of MCFA by 14 fold in the shake flask and 30 fold in a bioreactor. In sum, the authors constructed circuits in *E. coli* based on quorum-sensing and *mazF* global regulator that can increase gene expression significantly. Here are some questions and concerns:

- There are numerous typos and grammar errors in the manuscript. The English needs to be improved to achieve publishable quality. In addition there are many figures, many of which are not necessary, and thus the article needs to be condensed.
- For example, the microscopic pictures in Figure 4D are redundant with the plate reader data. Quantitative analysis of microscopic image is needed to compare to the fluorescence of single cells across different images and statistical analysis is needed.
- The statement that no orthogonal quorum-sensing systems were constructed in *E. coli* is not correct. See "Emergent genetic oscillations in a synthetic microbial consortium" by Chen et al. In this paper, they constructed an orthogonal quorum-sensing system in *E. coli* and achieved dual-strain oscillation.
- It is unclear why varying the timing of gene expression for both *mazF* and output genes via quorum-sensing can lead to increased gene expression, especially compared to a standard inducible gene expression system. Does this strategy minimize toxicity? Could experiments be designed compare to an inducible system and understand the mechanisms that enhance output expression? Also, to achieve maximal output gene expression, the timing of *mazF* expression needs to be tuned to be later than the output genes. What are the possible explanations and experiments that could be performed to understand this result?
- It is also unclear why the introduction of protected host factors can lead to further increase in output gene expression. Even the RNA of native genes will be degraded by *mazF* eventually, these protected host factors could still activate native genes, reducing the resource redistribution activity. What are the possible explanations and could experiments be performed to understand the mechanisms that lead to enhanced output expression in strains where the host factors are protected? For example, qPCR or RNA-seq could be useful to understand how protection of these genes impacts global gene expression patterns.
- The stability of the system was only examined for 60 hours. Evolutionary escape mutants could arise over longer time scales due to the metabolic burden of expression of many heterologous genes, including the toxic gene *mazF*. For metabolic engineering applications, the evolutionary stability of the system should be examined over a longer time window.

Reviewer #1 (Remarks to the Author):

"Developing a pathway-independent and full-autonomous global resource allocation strategy to dynamically switch phenotypic states" by Wu and colleagues describes the use of quorum sensing circuits to change the resource allocation between host and synthetic circuits during regulation. They first demonstrate this with a fluorescent reporter, but also show a metabolic engineering application of the production of fatty acids. The system described is a moderate advance on the work of the Prather lab where a quorum sensing system was used to direct flux from glycolysis for the production of myo-inositol and glucaric acid (Gupta et al, 2017, Nature Biotech, 35:273-279) in that it both redirects flux and also induces the expression of the synthetic circuit by using two orthogonal quorum regulators. It is also a minor advance on the work of Adam Arkin's group who already showed that MazF could be used for global resource allocation in E. coli when expressed (Venturelli et al, 2017, Nature Comms, 8, Article number: 15128). In the present case, MazF is autoinduced by internally synthesized AHL and the effect of recoding key host genes needed for expression (RNA polymerase, TFs, etc) is investigated.

Q1: My major criticism of this paper is its sheer size. At 126 pages, the submitted version is almost the length of a PhD thesis. There are 11 figures and 8 tables in the main manuscript, which is well above the number allowed by the journal. The tables, especially, do not add anything to the main text and could have been part of the supplementary information. I would have liked

to see the authors take the time to craft a clear, concise message from the data before submitting it for review. I understand that a lot of time and effort went into doing these experiments, but the temptation to show it all in the main text should be avoided. More is not always better.

A: Thanks for the suggestions. The whole manuscript has been condensed as follows, and now the manuscript contained 53 pages.

1) As the tables in the manuscript mainly provided the method information, the eight tables have been moved to the Supplementary Materials. **See Table S3-S11 in the Supplementary Materials.**

2) The part of Materials and methods has been condensed. The detailed information about construction of QS systems in *E. coli*, generation of different variants of PluxI promoters, construction of QS components tagged with different strength promoters, construction of global resource allocators, protection of hos factors including transcriptional and translational machinery, construction of resource allocator circuits to enhancing MCFA production yields, has been moved to Supplementary Materials. **See Page 30, Lines 582-584:** The detail information about construction of QS systems, different variants of PluxI promoters, global resource allocators, and protection of host factors was shown in supplementary methods in Supplementary Materials.

3) The figures have been condensed. Fig. 4D has been removed and replaced by Table S1 and S2. Fig. 7 has been removed to the Supplementary Materials as Fig. S3. **See Tables S1 and S2, Fig. S3 in Supplementary Materials.**

Q2: The English could use minor editing in places for subject-verb agreement, correct use of adjectives and adverbs, etc.

A: Minor editing has been used in the manuscript, and the grammar errors throughout the manuscript have been checked and corrected. All of these corrections are shown in color highlighting throughout the manuscript.

Q3: Page 7, lines 123-125, The authors say, "Although there are a few pioneering studies characterizing different QS systems to investigate orthogonal pairs (ref 11-13), no more than one type of QS system could be co-expressed in a single cell." The reference list here is incomplete as there have been more than a few studies in this area over the last few years.

A: This sentence has been corrected and other researches related to characterizing QS systems has been added. **See Page 7, Lines 123-126:** Although there are numerous pioneering studies characterizing different QS systems to investigate orthogonal pairs¹⁻⁸, the available QS systems that could be functional in a single cell without cross talk were still limited.

Q4: In addition, it is not clear what the authors mean by, "no more than one type of QS system could be co-expressed". This is phrased as to mean it is not possible, but maybe the authors mean it has not been attempted?

A: 1) A lot of pioneering studies have made significant advances in characterizing

orthogonal QS systems without cross talk. Some exciting studies focused on characterizing Lux-like QS systems from gram-negative bacterium^{1,2}, each with a unique LuxR-like receptor and homoserine lactone (HSL) homologue, and found that hybrid tra and rpa QS systems, which were created by replacing lux-box-like sequences from Lux QS systems with tra-box and rpa-box, respectively, exhibited both signal and promoter orthogonality². However, this hybrid characteristic often led to off-target binding of regulator proteins toward motifs of lux box promoter when co-expressed in a single cell¹. Some exciting studies identified the γ -butyrolactone system from *Streptomyces coelicolor* exhibited signal and promoter cross talk with quorum sensing systems from *Vibrio fischeri*. However, additional supplement of signal molecules into culture medium were required to activate gene circuits of γ -butyrolactone system in *E. coli*⁷. Another pioneering work constructed an orthogonal quorum-sensing system in two different *E. coli* strains and achieved dual-strain oscillation. Hence, the availability of QS systems that could be functional in a single cell was limited.

2) This sentence has been changed to make it more comprehensible. **See Page 7, Lines 123-126:** Although there are numerous pioneering studies characterizing different QS systems to investigate orthogonal pairs¹⁻⁸, the available QS systems that could be functional in a single cell without cross talk were still limited.

Q4: Page 10, line 171, referencing three of their own papers to discuss a three

plasmid system that has three different origins of replication and three different selection markers seems gratuitous.

A: Thanks for the suggestions. Two of our own papers have been deleted. **See Page 10, Lines 170-175:** To facilitate examining the interactions of different QS components, a modular three-plasmid system constructed in our previous studies, which could be stably maintaining together in each cell due to their different origins and resistance marker⁹, is used. Such that a signal plasmid (Ps) and a regulator plasmid (Pr) produced auto-inducers and transcription regulators under their native promoters, respectively, while a reportorial plasmid expressed GFP under a single QS-promoter (Pp) (Fig. 2A).

Q5: Page 10, lines 176-8, The authors say they compared the strains with three plasmids to a control that just had the reporter plasmid. This is not a properly controlled experiment because of the effects of maintaining multiple plasmids and synthesis of three antibiotic resistance markers on the cells. These studies should have been conducted using the reporter plasmid and two empty plasmid backbones corresponding to the Ps and Pa plasmids.

A: 1) Thanks for the suggestions. That is what we exactly conduct in this manuscript, and all the control experiment was performed with corresponding empty plasmids. In figure captions, we have already illustrated the detailed information. For example, at the end of figure caption of Fig. 2, we have illustrated the meaning of (-). **See Page 44, Lines 832-833:** (-) indicated the wild

type strain *E. coli* MG1655 with corresponding empty plasmids as the negative control. And also we illustrated this at the caption of Fig. 3, **See Page 45, Lines 838-841**: The induction ratio was defined as dividing the normalized GFP fluorescence from different combinations by the control strain harboring only response promoter and two corresponding empty plasmids.

2) We apologize for the unclear explanation in the manuscript and this sentence has been corrected. **See Page 10, Lines 177-179**: The function of each QS system was validated by comparing normalized GFP fluorescence from strains harboring plasmids of Ps, Pa and Pp to strains only harboring Pp and corresponding empty plasmids.

Q6: While normalizing GFP fluorescence by the OD600 can control for small differences in growth rates, the fluorescence/OD is not perfectly linear due to the effects of light scattering on fluorescence measurements.

A: Thanks for the comment. In this study, we normalized GFP fluorescence by the OD₆₀₀ in order to conduct a per-cell measurement based on previous work¹⁰. In order to minimize the deviation, all the samples have been conducted on the same machine and using the same method. However, more precise and advance method to fix the effects of light scattering on fluorescence measurements is definitely required and need to be developed in future work.

Q6: Bar charts should show individual data points as well as the mean and

standard deviation

A: All the individual data points associate with the mean and standard deviation are provided in Source Data file.

Q7: Figure 2D, where are three bars for the PprgQ system? Is this to show that it is constitutive in the absence of its regulator?

A: Yes, the three bars for PprgQ system were used to show the promoter strength in the absence of its regulator. In contrast with AHL-based QS systems that employ positive-feedback control mechanisms, prgX QS system adopts different control mechanisms. Master protein regulator PrgX binds the operator sequence of the response promoter from prgQ operon and represses the expression of this operon. Once bound to heptapeptide cCF10 produced by heptapeptide synthase CcfA, a conformational change would occur to the protein regulator PrgX and could no longer interact with the operator sequence, thus led to the activation of prgQ operon. Hence, in this part, we used three bars to show whether the promoter of prgQ operon is constitutive, and its expression strength in the absence of PrgX.

Q8: Lines 267 ff, 'leftward' and 'rightward' operon don't seem particularly technical terms.

A: 1) 'leftward' and 'rightward' have been corrected as 'upstream side' and 'downstream side', respectively.

2) **See Page 15, Lines 268-270:** When deleting from downstream side of the

operon, the leakiness level of PluxI promoter decreased dramatically (I1-I3), which was measured in the absence of signal inducer and actuator (Fig. 4B).

3) See Page 15, Lines 273-275: Whereas deleting from upstream side of the operon, the leakiness level remained unchanged until 69 bp from leftward operon (I4-I7), and commenced dramatically decreasing from 69 bp to 109 bp (I7-I9).

4) See Page 46, Lines 852-856: Fig. 4 Rewiring QVX-controlled gene expression mechanism from positive feedback loop to on-time transcription circuit for decreasing leakiness. (A) Schematic of deletion mapping of regulatory region between LuxR and LuxI. I1-I3 engineered promoters were deleted from downstream side of regulatory region between LuxR and LuxI, while I4-I12 from upstream side of regulatory region.

Q9: Why are the mKate samples in Figure 7 labelled as 'EFP'?

A: 'EFP' in Figure 7 has been corrected as 'mKate2'. **See Figure S3 in Supplementary Materials.**

Q10: I think you mean 'recoded' where you have said 'recorded'

A: Yes. 'recorded' in the manuscript has been corrected as 'recoded'. **See Page 20, Lines 381-383:** As auto-inducer synthase and transcriptional regulator of QS systems are enriched for MazF recognition sites (Gene sequences shown in Supplementary materials), these four proteins (LuxI, LuxR and CcfA, PrgX) were recoded to remove these sites.

Q11: Is the data in Figure 8B representative curves (n=1) or were replicate measurements taken? It would be helpful to include error bars to understand the reproducibility of various autoinduction conditions as well as whether individual conditions are different from each other

A: 1) As shown in Materials and methods, each experiment in the manuscript was conducted in triplicate and the averaged values were used for analysis. **See Page 33, Lines 629-631:** For plate reader experiments, each experiment was performed in triplicate through inoculating into different wells in 96-well plates and the averaged values were used for analysis.

2) As there are seven curves in each graph, previously we were afraid that including error bars in the graph may make the graph unclear and mix the trend of each curve. However, it is very important to understand the reproducibility of different conditions directly, and the error bars have been included in the graph. **See Fig. 7B.**

Fig. 7 Design of a pathway-independent and full-autonomous global resource allocation strategy. (A) Comparing normalized fluorescence of GFP-N and GFP-M driven by QVX systems. PluxI I11 indicated engineered PluxI promoter of I11. QVX Pi GFP(M) or QVX Pi GFP(N) indicated normalized fluorescence of GFP(M) or GFP(N) driven by QVX system using Pi promoter for LuxR and LuxI expression (i=1-6). (B) The impact of different triggering times of MazF expression on fluorescence dynamics driven by QVX system

using P1-P6 promoters (B1-B6) for LuxR and LuxI expression. QEX Pi MazF indicated introducing MazF driven by QEX circuit using Pi promoter for CcfA expression into control strains harboring QVX Pi GFP(M) circuits (i=1-6). (C) The impact of MazF expression on fluorescence fold change. (D) The effect of protecting QS system components from MazF on fluorescence fold change. QVX P2 GFP(M) indicated GFP(N) expression driven by QVX system using P2 promoter for LuxR and LuxI expression. QVX P2 GFP(M), QEX P3 MazF indicated introducing QEX P3 MazF circuit into strains harboring QVX P2 GFP(M) circuit. LuxI(M), LuxR(M) indicated replacing LuxI and LuxR with LuxI(M) and LuxR(M) in strains harboring QVX P2 GFP(M), QEX P3 MazF. CcfA(M), PrgX(M), PrgZ(M) indicated replacing CcfA, PrgX and PrgZ with CcfA(M), PrgX(M) and PrgZ(M) in strains harboring QVX P2 GFP(M), QEX P3 MazF. LuxI(M), LuxR(M), CcfA(M), PrgX(M), PrgZ(M) indicated replacing LuxI, LuxR, CcfA, PrgX and PrgZ with LuxI(M), LuxR(M), CcfA(M), PrgX(M), PrgZ(M) in strains harboring QVX P2 GFP(M), QEX P3 MazF. Pre-cultured recombinant cells were diluted to OD₆₀₀ of 0.01 in 96-well plate in 200 µL of MOPS medium on an Infinite M1000 PRO (Tecan, Switzerland) plate reader at 30 °C. Cell density and fluorescence were measured every 1 h for 30 h.

3) All the individual data points associate with the mean and standard deviation were also provided in Source Data file to show the reproducibility of various experiment conditions. .

Q12: Page 25, line 480. I would suggest changing the title of this section. You have not really investigated the stability of the circuit, but rather its performance under industrial conditions.

A: Thanks for the suggestion. The title of this section has been changed as ‘Evaluating the resource allocator circuit performance in bioreactors’. **See Page 25, Line 483.**

Q13: I also do not understand the reason why overflow metabolites are presented as part of this study.

A: The overflow metabolites from different engineered strains were measured to demonstrate the redistribution of cellular byproducts toward synthetic circuits due to the implementation of resource allocator circuit. Hence, the concentrations of four main byproducts such as acetate, ethanol, lactate, and succinate, were presented in this part. It was found that these byproduct concentrations decreased dramatically when implementing the resource allocators, further demonstrating the functionality of resource allocator circuit.

Reviewer #2 (Remarks to the Author):

The authors tested four quorum-sensing systems from different bacteria and found that the quorum-sensing systems from *Vibrio fischeri* and *Enterococcus faecalis* can be further engineered to achieve tight and orthogonal control of gene expression in *E. coli*. The timing of the gene expression of the quorum systems can be further fine-tuned by varying the expression of regulators, which allows differential gene expression during cellular growth. They used the endoribonuclease *mazF* to divert cellular resources to the output gene expression and enhance GFP or target molecules, by protecting genes from *mazF*-mediated RNA degradation. The output expression was significantly enhanced when both the *mazF* and output genes are expressed from the quorum-sensing systems, with specific fine-tuned timing. To further increase the output gene expression, the authors protected the regulators of the quorum-sensing circuit and also the transcriptional and translational machinery. Using the same system, they introduced four metabolic genes for the reverse beta oxidation pathway and demonstrated an increase of MCFA by 14 fold in the shake flask and 30 fold in a bioreactor. In sum, the authors constructed circuits in *E. coli* based on quorum-sensing and *mazF* global regulator that can increase gene expression significantly. Here are some questions and concerns:

Q1: There are numerous typos and grammar errors in the manuscript. The English needs to be improved to achieve publishable quality.

A: The typos and grammar errors throughout the manuscript have been checked and corrected. The color highlighting is used to show the corrections of these errors.

Q2: In addition there are many figures, many of which are not necessary, and thus the article needs to be condensed.

A: The whole manuscript including tables, materials and methods, figures, have been condensed as follows, and now the manuscript contained 53 pages.

1) As the tables in the manuscript mainly provided the method information, the eight tables have been moved to the Supplementary Materials. **See Table S3-S11 in the Supplementary Materials.**

2) The part of Materials and methods has been condensed. The detailed information about construction of QS systems in *E. coli*, generation of different variants of PluxI promoters, construction of QS components tagged with different strength promoters, construction of global resource allocators, protection of hos factors including transcriptional and translational machinery, construction of resource allocator circuits to enhancing MCFA production yields, has been moved to Supplementary Materials. **See Page 30, Lines 582-584:** The detail information about construction of QS systems, different variants of PluxI promoters, global resource allocators, and protection of host factors was shown in supplementary methods in Supplementary Materials.

3) The figures have been condensed. Fig. 4D has been removed and replaced by Table S1 and S2. Fig. 7 has been removed to the Supplementary Materials as Fig. S3. **See Table S1 and S2, Fig. S3 in Supplementary Materials.**

Q3: For example, the microscopic pictures in Figure 4D are redundant with the plate reader data. Quantitative analysis of microscopic image is needed to compare to the fluorescence of single cells across different images and statistical analysis is needed.

A: 1) Thanks for the suggestions. Quantitative analysis of microscopic image in Fig. 4D was conducted by ImageJ (<https://imagej.nih.gov/ij/>). The type of microscopic image was firstly converted to 8-bit, and the resulted image was then adjusted threshold for further analyzing the fluorescence of single cells. For each microscopic image, three different single cells were chose for further comparison except for I2, I3 and I12, as no fluorescence had been detected in these three images (Table S1).

Table S1. Quantitative analysis of microscopic image

	Fluorescence values (a.u.)*									
	Control	I1	I4	I5	I6	I7	I8	I9	I10	I11
Cell 1	21.71	6.97	17.91	18.08	15.58	11.05	5.58	4.56	4.33	3.66
Cell 2	16.21	7.78	17.05	16.27	15.34	10.82	5.46	5.58	4.01	3.84
Cell 3	18.66	8.44	20.63	15.75	20.72	11.46	5.87	5.37	4.28	3.63
Mean	18.86	7.73	18.53	16.71	17.21	11.11	5.64	5.17	4.21	3.71
Error bar	2.76	0.74	1.87	1.22	3.04	0.32	0.21	0.54	0.17	0.11

*a.u. means arbitrary unit.

2) Statistical analysis has been conducted by IBM SPSS Statistics analysis (Table. Xx). It was found that cell fluorescence in different images showed significant

difference ($p < 0.01$), and cell fluorescence in group of control, I4, I5, I6, or group of I8, I9, I10, I11 showed no significant difference ($p > 0.05$). This trend matches well with our previous conclusions: When deleting from downstream side of the operon, the leakiness level of PluxI promoter decreased dramatically (I1-I3) (Fig. 4B). Whereas deleting from upstream side of the operon, the leakiness level remained unchanged until 69 bp from leftward operon (I4-I7), and commenced dramatically decreasing from 69 bp to 109 bp (I7-I9). Only a slight decrease of leakiness level was observed through deleting from 109 bp to 139 bp (I9-I11).

Table S2. Statistical analysis of fluorescence values in different microscopic images

	Control	I4	I5	I6	I11	I7	I8	I9	I10	I11
P value in different images	0.00006									
P value in groups	0.326					0.423				

Q4: The statement that no orthogonal quorum-sensing systems were constructed in E. coli is not correct. See “Emergent genetic oscillations in a synthetic microbial consortium” by Chen et al. In this paper, they constructed an orthogonal quorum-sensing system in E. coli and achieved dual-strain oscillation.

A: Thanks for the suggestions. This sentence has been changed and this paper has been cited in the manuscript. **See Page 7, Lines 123-126:** Although there are numerous pioneering studies characterizing different QS systems to investigate

orthogonal pairs¹⁻⁸, the available QS systems which could be functional in a single cell without cross-talk were still limited.

Q5: It is unclear why varying the timing of gene expression for both *mazF* and output genes via quorum-sensing can lead to increased gene expression, especially compared to a standard inducible gene expression system. Does this strategy minimize toxicity? Could experiments be designed compare to an inducible system and understand the mechanisms that enhance output expression?

A: 1) Yes, in this study, in order to improve the behavior of target synthetic circuits, we need to vary the timing of gene expression for both *mazF* and output genes to minimize both toxicity and metabolic burden, which were associated with heterogenous gene expression and the resource allocation capability of MazF.

2) The expression of the output genes would impose a metabolic burden on the host strain, which came from expression of plasmid-borne resistance, replication of the plasmids, and QS components. Therefore, an early triggering time point would lead to significant phenotypic effects, such as growth retardation. Hence, it would be better for cells to firstly thrive under stringent media conditions before imposing metabolic burden associated with protein overexpression. However, a delay triggering time point would channel more metabolic flux into biomass accumulation. Both of these cases would lead to sub-optimal gene expression. Therefore, a suitable triggering time point is vital to achieve the best

gene expression, and expression of different genes or pathways would require different triggering time points. For example, in our previous study for flavonoid production¹¹, a little delay triggering time point would alleviate the metabolic burden and achieve the highest production yield, while in the case of mCherry expression¹⁰, delayed induction by 2 h would reduce mCherry expression by 85%. Thus, an early or delay triggering time point for target synthetic circuits would decrease their behaviors, and suitable timing of gene expression need to be explored in each case.

3) There is one experiment to further explain this phenomenon. As seen from Fig. 6A, the peak GFP fluorescence of QVX P1 was lower comparing with QVX P2, while the triggering time of QVX P1 was earlier than QVX P2. This may be due to excessively expressed QS components and earlier triggering time points, both of which imposed unexpected metabolic burden to the cell, as the hampered cellular growth of engineered strains harboring QVX P1 circuit was observed (Fig. S2). However, in the case of mKate2 expression under different QEX components (Fig. 6B), we did not observe the same phenomenon, further demonstrating expression of different genes under different circumstances would require different triggering time points. Hence, in this manuscript for expressing different genes, we always varied the timing of gene expression to achieve the best performance.

Fig. 6 Identifying orthogonal QS systems functional in a single cell. (A)

Examining individual QVX response behavior with different strength promoters

for QS component expression. Response promoter PluxI tagged with GFP was placed on pCOLADuet-1. LuxR and LuxI were expressed under different strength promoters (P1-P6) on pACYCDuet-1. (B) Examining individual QEX response behavior with different strength promoters for CcfA expression. Response promoter PprgQ tagged with mKate2 was placed on pCOLADuet-1. PrgX and PrgZ were placed under P1 and Ptrc constitutive promoter, respectively, on pACYCDuet-1, while CcfA was placed under Pi (i=1-6) promoter on the same plasmid. (C) Assembling QVX and QEX systems into a single cell to test their response functions. For each graph, left coordinates indicated normalized GFP fluorescence, while right one indicated normalized mKate2 fluorescence. All the QS components were placed on pACYCDuet-1, while response promoter PluxI and PprgQ tagged with GFP and mKate2 were placed on pCOLADuet-1. QVX P1 indicated LuxR and LuxI were expressed under P1 promoter, while QEX P1 indicated CcfA was expressed under P1 promoter. Assemble QVX indicated QVX in assemble systems, while single QVX indicated single QS system. Pre-cultured recombinant cells were diluted to OD₆₀₀ of 0.01 in 96-well plate in 200 µL of MOPS medium on an Infinite M1000 PRO (Tecan, Switzerland) plate reader at 30 °C. Cell density and fluorescence were measured every 1 h for 30 h.

Fig. S2 Cell growth (final OD₆₀₀) of engineered strains used different strength promoters to drive QVX component expression. This figure showed final OD₆₀₀ of different engineered strains using P1-P6 promoters to express LuxR and LuxI. The recombinant strains were grown in 25 mL of MOPS

medium at 30 °C with 220 rpm orbital shaking. Cell density (OD_{600}) was measured after 30 h of culture on a Cytation 3 imaging reader system (BioTek, Winooski, USA). For each measurement, triplicate cultures are conducted and an error bar is utilized to represent their deviation.

4) Except for metabolic burden associated with heterogenous gene expression, the timing of MazF need to be varied due to its capability of resource allocation. The early triggering of MazF might dramatically shorten normal growth period, thus limiting cellular resources for minimum cell growth, as the final OD_{600} decreased dramatically by 87.1% (Fig. S4). Whereas the delay triggering cannot efficiently prevent cellular resources from cell growth toward target functions. As shown in Fig. 7B, it was found that, following MazF induction, different triggering times of QEX circuits driving MazF expression imposed different impact on fluorescence dynamics (Fig. 7B). A suitable triggering time of QEX

P3 MazF circuit acting on QVX P2 GFP circuit presented the best performance (approximately 3-fold increase in fold change), while an early or delay triggering time points would lead to a decrease in fold-change (Fig. 7C). Hence, the timing of gene expression for both MazF and output genes need to be investigated in each case to find the suitable timing to balance cellular resources between cell growth and target synthetic circuit.

Fig. S4 The impact of different triggering times of MazF expression on cell growths. This figure presented final OD₆₀₀ of engineered strains with different triggering times of MazF expression. The engineered strains harbored QVX P2 GFP(M) and QEX Pj MazF circuits (j=1-6). QVX P2 GFP(M) indicated GFP(M) driven by QVX circuit using P2 promoter for LuxR and LuxI expression. QEX Pj MazF indicated MazF driven by QEX circuit using Pj promoter for CcfA expression. The recombinant strains were grown in 25 mL of MOPS medium at 30 °C with 220 rpm orbital shaking. Cell density (OD₆₀₀) were measured after 30 h of culture on a Cytation 3 imaging reader system (BioTek, Winooski, USA). Three biological replicates were performed and averaged for each sample.

Fig. 7 Design of a pathway-independent and full-autonomous global

resource allocation strategy. (A) Comparing normalized fluorescence of GFP-N and GFP-M driven by QVX systems. PluxI I11 indicated engineered PluxI promoter of I11. QVX Pi GFP(M) or QVX Pi GFP(N) indicated normalized fluorescence of GFP(M) or GFP(N) driven by QVX system using Pi promoter for LuxR and LuxI expression (i=1-6). (B) The impact of different triggering times of MazF expression on fluorescence dynamics driven by QVX system using P1-P6 promoters (B1-B6) for LuxR and LuxI expression. QEX Pi MazF indicated introducing MazF driven by QEX circuit using Pi promoter for CcfA expression into control strains harboring QVX Pi GFP(M) circuits (i=1-6). (C) The impact of MazF expression on fluorescence fold change. (D) The effect of protecting QS system components from MazF on fluorescence fold change. QVX P2 GFP(M) indicated GFP(N) expression driven by QVX system using P2 promoter for LuxR and LuxI expression. QVX P2 GFP(M), QEX P3 MazF indicated introducing QEX P3 MazF circuit into strains harboring QVX P2 GFP(M) circuit. LuxI(M), LuxR(M) indicated replacing LuxI and LuxR with LuxI(M) and LuxR(M) in strains harboring QVX P2 GFP(M), QEX P3 MazF. CcfA(M), PrgX(M), PrgZ(M) indicated replacing CcfA, PrgX and PrgZ with CcfA(M), PrgX(M) and PrgZ(M) in strains harboring QVX P2 GFP(M), QEX P3 MazF. LuxI(M), LuxR(M), CcfA(M), PrgX(M), PrgZ(M) indicated replacing LuxI, LuxR, CcfA, PrgX and PrgZ with LuxI(M), LuxR(M), CcfA(M), PrgX(M), PrgZ(M) in strains harboring QVX P2 GFP(M), QEX P3 MazF. Pre-cultured

recombinant cells were diluted to OD₆₀₀ of 0.01 in 96-well plate in 200 µL of MOPS medium on an Infinite M1000 PRO (Tecan, Switzerland) plate reader at 30 °C. Cell density and fluorescence were measured every 1 h for 30 h.

Q6: Also, to achieve maximal output gene expression, the timing of mazF expression needs to be tuned to be later than the output genes. What are the possible explanations and experiments that could be performed to understand this result?

A: 1) As shown in the previous question, the similar mechanism lies behind the result that the timing of MazF expression needs to be tuned to be later than the output genes, as that is the suitable timing to balance cellular resources between cell growth and target synthetic circuit. Following MazF induction, it was found that different triggering times of QEX circuits driving MazF expression imposed different impact on fluorescence dynamics (Fig. 7B). A suitable triggering time of QEX P3 MazF circuit acting on QVX P2 GFP circuit presented the best performance (approximately 3-fold increase in fold change), while an early or delay triggering time points would lead to a decrease in fold-change (Fig. 7C). The early triggering of MazF might dramatically shorten normal growth period, thus limiting cellular resources for minimum cell growth, as the final OD₆₀₀ decreased dramatically by 87.1% (Fig. S4). Whereas the delay triggering cannot efficiently prevent cellular resources from cell growth toward target functions.

2) In addition, as seen from Fig. 7B and 7C, different triggering time points of output genes required different timing of MazF expression to achieve the maximal output gene expression in different cases. In cases of B1, B2 in Fig. 7B, the timing of MazF expression needed to be tuned to be later than the output

genes to achieve the maximal expression, whereas in B5, B6 in Fig. 7B, an earlier timing of MazF expression was required to achieve the maximal output gene expression due to the late triggering time of output genes. This further demonstrated the importance of varying the timing of gene expression for both *mazF* and output genes to balance cellular resources between cell growth and target synthetic circuit.

Q7: It is also unclear why the introduction of protected host factors can lead to further increase in output gene expression. Even the RNA of native genes will be degraded by mazF eventually, these protected host factors could still activate native genes, reducing the resource redistribution activity. What are the possible explanations and could experiments be performed to understand the mechanisms that lead to enhanced output expression in strains where the host factors are protected? For example, qPCR or RNA-seq could be useful to understand how protection of these genes impacts global gene expression patterns.

A: 1) The output gene expression depends on a dense network of transcriptional and translational machinery and MazF-mediated decay of these factors would disrupt their expression. Hence, essential factors that could improve the resource allocator performance after protection from MazF need to be examined. As PluxI response promoter is a RNA polymerase-dependent promoter ³², and furthermore, previous proteome analysis found that three ribosomal protein subunits S9, S20,

L17 and elongation factor EF-Ts decreased dramatically¹, these factors were then examined in the manuscript. Although these protected host factors could also activate native genes, most native genes would be degraded by MazF and could not function as mature enzymes to channel carbon flows into byproducts. Whereas the target synthetic circuits have been protected by MazF, thus driving more carbon fluxes into synthetic circuits. This has been demonstrated in the manuscript. As seen from Fig. 10C, at the end of each fermentation (60 h), the concentrations of four main byproducts such as acetate, ethanol, lactate, succinate were measured. It was found that these byproduct concentrations decreased, and titers of target product MCFA increased when introducing the protected host factors (Fig. 10C2 and 10C3).

Fig. 10 Evaluating resource allocator performance in a 5-L fermenter. (A) The compositions of S_B1, S_B2 and S_B3. S_B1 consisted of pathway module and signal module as control. S_B2 and S_B3 consisted of three modules, while previous identified host factors were protected from MazF on S_B3. (B) Time course of total MCFA production (B1), cell growth (B2), and total glucose consumption (B3) of S_B1, S_B2 and S_B3 in a 5-L fermenter. (C) Time course of four main byproducts accumulation of S_B1 (C1), S_B2 (C2) and S_B3 (C3) in a 5-L fermenter. (D) The comparison of MCFA production among S_B1, S_B2 and S_B3.

D

Strain	MCFA titer (g/L)	Final OD ₆₀₀	Specific titer (g/L/OD ₆₀₀)	Yield (g MCFA/g glucose)
S_B1	0.76	18.06	0.042	0.006
S_B2	1.9	4.31	0.441	0.116
S_B3	4.9	4.5	1.089	0.187

2) In order to further understand the mechanisms, qPCR was conducted to compare the mRNA levels between synthetic circuits and native byproduct pathways at the stationary phase when introducing protected host factors into our final engineered strains. Native genes encoding enzymes responsible for

synthesizing main by-products such as ethanol (*adhE*), acetate (*pta* and *poxB*), lactate (*ldhA*) and succinate (*frdA*) were chosen for qPCR measurements. It was found that mRNA levels of synthetic circuit genes such as *bktB*, *fadB*, *ter*, *ydiI*, increased dramatically after introducing protected host factors, whereas no significant change was observed among native byproduct pathway genes. This further demonstrated that protection of host factors was vital for improving the behavior of target synthetic circuits.

Fig. S10 qPCR measurements to investigate change patterns of mRNA levels between target synthetic circuit and native byproduct pathway. qPCR was conducted to compare the mRNA levels between synthetic circuits and native byproduct pathways at the stationary phase when introducing protected host factors into our final engineered strains. The reference gene *cysG*, which did not change significantly in response to MazF activity based on previous RNA-seq experiments⁹, was used to normalize qPCR values with primers Pf_*cysG*/Pr_*cysG* for amplifying. The recombinant strains were grown in 50 mL of MOPS medium at 30 °C with 220 rpm orbital shaking, and cells were collected at the stationary phase. Three biological replicates were performed and averaged for each sample.

3) However, even the RNA of native genes will be degraded by MazF eventually and most of them could not function as mature enzymes to channel carbon flows into byproducts, the activation of native genes due to these protected host factors could still sequester resources from engineered circuits. Hence, there would be a balance between the extent of protecting host factors and the sacrifice of cellular resources. Therefore, in this study, although protecting EF-Ts, S9, S20, L17 individually exhibited improved performance on resource redistribution activity, a combination of these protected factors produced no observable stacked effect, indicating that the suitable protection extent yielded the highest redistribution activity.

4) In future study, in order to completely solve the problem, different orthogonal RNA polymerases, such as T7 RNA polymerases, could be developed to drive engineered circuits while at the same time totally inactivating native RNA polymerases, which could minimize the waste of energy and carbon flows.

Q8: The stability of the system was only examined for 60 hours. Evolutionary escape mutants could arise over longer time scales due to the metabolic burden of expression of many heterologous genes, including the toxic gene mazF. For metabolic engineering applications, the evolutionary stability of the system should be examined over a longer time window.

A: 1) Thanks for the suggestions. As shown in Fig. 10B, we only examined the stability of the system for 60 hours in bioreactors because the actual fermentation time of final engineered strain S_B3 was approximately only 55 hours, and for other two control strains, such as S_B1 and S_B2, the actual fermentation time was more shorter (approximately 40 hours). After the actual fermentation time, the strain performance maintained stable and the target product titer remained constant. For example, as seen from Fig. 10B1, the performance of S_B2 and S_B3 remained stable from the end of actual fermentation time (approximately 40 h) to 60 h. Even the stability of the system was unstable after 60 h, it exerted little effect on strain performance, as we had already shut down the bioreactor and started to extract the target products.

2) However, as the reviewer pointed out, for metabolic engineering applications, the evolutionary stability of the system was very important. In our previous experience, we found that these engineered strains are very stable in shake flasks and bioreactors in the first five generations (at least five generations) during the fermentation time after transforming corresponding plasmids. After the first five

generations, the titers may decrease about 30% compared to the first generations due to strain degeneration. This issue would be overcome after we transformed corresponding plasmids again. Hence, in this manuscript, we would investigate the effect of different engineering strategies on MCFA titers in the first three generations after transforming corresponding plasmids. The transforming of corresponding plasmids would be conducted again when the generation numbers exceed three. In the future study, some pathway essential genes would be integrated into the chromosome of host strain to minimize the usage of plasmids and improve the stability of engineered strains.

References

1. Tekel, S.J. et al. Engineered Orthogonal Quorum Sensing Systems for Synthetic Gene Regulation in *Escherichia coli*. *Frontiers in Bioengineering and Biotechnology* **7**, 12 (2019).
2. Scott, S.R. & Hasty, J. Quorum Sensing Communication Modules for Microbial Consortia. *ACS. Synth. Biol* **5**, 969-977 (2016).
3. Halleran, A.D. & Murray, R.M. Cell-Free and In Vivo Characterization of Lux, Las, and Rpa Quorum Activation Systems in *E-coli*. *ACS. Synth. Biol* **7**, 752-755 (2018).
4. Scott, S.R. et al. A stabilized microbial ecosystem of self-limiting bacteria using synthetic quorum-regulated. *Nat. Microbiol* **2**, 9 (2017).
5. Chen, Y., Kim, J.K., Hirning, A.J., Josić, K. & Bennett, M.R. Emergent genetic oscillations in a synthetic microbial consortium. *Science* **349**, 986-989 (2015).
6. Silva, K.P.T., Chellamuthu, P. & Boedicker, J.Q. Quantifying the strength of quorum sensing crosstalk within microbial communities. *PLoS Comp. Biol.* **13**, e1005809 (2017).
7. Biarnes-Carrera, M., Lee, C.K., Nihira, T., Breitling, R. & Takano, E. Orthogonal Regulatory Circuits for *Escherichia coli* Based on the gamma-Butyrolactone System of *Streptomyces coelicolor*. *ACS. Synth. Biol* **7**, 1043-1055 (2018).
8. Dinh, C.V. & Prather, K.L.J. Development of an autonomous and bifunctional

quorum-sensing circuit for metabolic flux control in engineered *Escherichia coli*. *Proc Natl Acad Sci U S A* **116**, 25562-25568 (2019).

9. (!!! INVALID CITATION !!!).
10. Venturelli, O.S. et al. Programming mRNA decay to modulate synthetic circuit resource allocation. *Nat. Commun.* **8**, 11 (2017).
11. Wu, J.J., Du, G.C., Zhou, J.W. & Chen, J. Metabolic engineering of *Escherichia coli* for (2S)-pinocembrin production from glucose by a modular metabolic strategy. *Metab. Eng.* **16**, 48-55 (2013).

Reviewers' Comments:

Reviewer #1:

Remarks to the Author:

Overall, the authors have addressed the majority of my comments to my satisfaction. The streamlined manuscript is much more tractable to read.

There are still a few minor problems with the English that could be resolved.

Just a few examples from the abstract:

" over fermentation course" should be "over the course of the fermentation"

"only allow for a single QS circuit functional in one cell" should be "only have capacity for a single QS circuit..."

"was developing " should be "was developed"

"increase of" should be "increase in" etc.

Perhaps this is something that will be fixed by the journal at typesetting. If not, the authors should consider employing a manuscript editing service.

Reviewer #3:

Remarks to the Author:

The authors have addressed/replied to previous raised concerns.

There are still a couple of criticisms:

1. The manuscript is still too long. The message should be concise, otherwise the reader get lost in the number of test performed. I suggest to cut the number of figures of the main text to maximum 5, to highlight the most interesting characterisation/outcome.

Also the text should be drastically cut and the exceeding parts moved as supplementary information. In particular I would focus on the systems that worked better and orthogonally and remove all the others from the main.

2. Despite grammar checks there are still typos. For example in the abstract line 33 (developing should be developed, redistributing should be redistribute etc). Also when mentioning in the abstract MCFA, the extended term should be mentioned

Reviewer #1 (Remarks to the Author):

Overall, the authors have addressed the majority of my comments to my satisfaction. The streamlined manuscript is much more tractable to read. There are still a few minor problems with the English that could be resolved. Just a few examples from the abstract:

"over fermentation course" should be "over the course of the fermentation";

"only allow for a single QS circuit functional in one cell" should be "only have capacity for a single QS circuit...";

"was developing" should be "was developed";

"increase of" should be "increase in" etc.

Perhaps this is something that will be fixed by the journal at typesetting. If not, the authors should consider employing a manuscript editing service.

Q1: "over fermentation course" should be "over the course of the fermentation";

A: This issue has been corrected. See **Page 2, Lines 23-25:** Quorum sensing (QS)-based dynamic pathway regulations provide a pathway-independent way to rebalance metabolic flux over the course of the fermentation.

Q2: "only allow for a single QS circuit functional in one cell" should be "only have capacity for a single QS circuit...";

A: This issue has been corrected. See **Page 2, Lines 25-27:** Most cases, however, these pathway-independent strategies only have capacity for a single QS circuit

functional in one cell.

Q3: "was developing" should be "was developed";

A: This issue has been corrected. **See Page 2, Lines 28-31:** Here, with the aid of engineering synthetic orthogonal quorum-related circuits and global mRNA decay, we report a pathway-independent dynamic resource allocation strategy, which allows us to independently controlling two different phenotypic states to globally redistribute cellular resources toward synthetic circuits.

Q4: "increase of" should be "increase in" etc.

A: **The same issue throughout the manuscript has been corrected. See Page 23, lines 445-451:** The implementation of this strategy in *E. coli* led to a 23.6-fold increase in reporter gene fluorescence (Fig. 5), and its application in biotechnological processes was further extending to MCFA production in both shake flask culture and 5-L bioreactors. This resulted in a 14.2-fold increase in specific MCFA titer in shake flask cultures (Fig. 6) and a 24.9- and 30.2-fold increase in specific MCFA titer and yield in bioreactors (Fig. S13), demonstrating the broad utility and large-scale application. **See Page 25, lines 488-491:** Here, it was showed that coupling programmable mRNA decay to dynamic regulation mechanism, cellular resources could be reallocating into target pathways with the protection of engineered network and host factors, led to a 14.2- and 24.9-fold increase in specific MCFA titer in shake flask cultures

(Fig. 6) and bioreactors (Fig. S13).

Q5: Perhaps this is something that will be fixed by the journal at typesetting. If not, the authors should consider employing a manuscript editing service.

A: All the similar issue has been corrected throughout the manuscript.

Reviewer #3 (Remarks to the Author):

The authors have addressed/replied to previous raised concerns.

There are still a couple of criticisms:

1. The manuscript is still too long. The message should be concise, otherwise the reader get lost in the number of test performed. I suggest to cut the number of figures of the main text to maximum 5, to highlight the most interesting characterization/outcome.

Also the text should be drastically cut and the exceeding parts moved as supplementary information. In particular I would focus on the systems that worked better and orthogonally and remove all the others from the main.

2. Despite grammar checks there are still typos. For example in the abstract line 33 (developing should be developed, redistributing should be redistribute etc).

Also when mentioning in the abstract MCFA, the extended term should be mentioned

Q1: The manuscript is still too long. The message should be concise, otherwise

the reader get lost in the number of test performed. I suggest to cut the number of figures of the main text to maximum 5, to highlight the most interesting characterization/outcome.

A: Thanks for the suggestion. The number of figures of the main text was cut to 6. The remaining six figures included the most important outcome from characterization to application of the constructed system.

Q2: Also the text should be drastically cut and the exceeding parts moved as supplementary information. In particular I would focus on the systems that worked better and orthogonally and remove all the others from the main.

A: The main text has also been cut. The parts of identifying orthogonal QS systems functional in a single cell and enhancing resource redistribution activity by protecting transcriptional and translational machinery have been moved as supplementary results in Supplementary materials.

Q3: Despite grammar checks there are still typos. For example in the abstract line 33 (developing should be developed, redistributing should be redistribute etc). Also when mentioning in the abstract MCFA, the extended term should be mentioned

A: All the issues have been corrected in the manuscript. Due to the limitation of 150 words in the abstract, these sentences have been deleted. However, we have checked the overall manuscript very carefully to avoid any same issue. The first

appearance of MCFAs was changed to the extended term. **See Page 2, lines 102-105:** As a demonstration, this dynamic resource allocation controller was implemented to control the production of medium chain fatty acids (MCFAs) in both shake flask cultures and 5-L bioreactors, exhibiting the potential of large-scale application.